palaeontology/evolution

Cambrian explosion, Burgess Shale, radiodont

**Author for correspondence:**
J.-B. Caron
e-mail: jcaron@rom.on.ca

# A giant nektobenthic radiodont from the Burgess Shale and the significance of hurdiid carapace diversity

J.-B. Caron[1,2,3] and J. Moysiuk[1,2]

[1]Department of Natural History, Royal Ontario Museum, 100 Queen's Park, Toronto, Ontario M5S 2C6, Canada
[2]Department of Ecology and Evolutionary Biology, University of Toronto, 25 Willcocks Street, Toronto, Ontario M5S 3B2, Canada
[3]Department of Earth Sciences, University of Toronto, 22 Russell Street, Toronto, Ontario M5S 3B1, Canada

J-BC, 0000-0002-1670-5502; JM, 0000-0002-4685-5819

Radiodonts, stem-group euarthropods that evolved during the Cambrian explosion, were among the largest and most diversified lower palaeozoic predators. These animals were widespread geographically, occupying a variety of ecological niches, from benthic foragers to nektonic suspension feeders and apex predators. Here, we describe the largest Cambrian hurdiid radiodont known so far, *Titanokorys gainesi*, gen. et sp. nov., from the Burgess Shale (Marble Canyon, Kootenay National Park, British Columbia). Estimated to reach half a metre in length, this new species bears a very large ovoid-shaped central carapace with distinct short posterolateral processes and an anterior spine. Geometric morphometric analyses highlight the high diversity of carapace shapes in hurdiids and show that *Titanokorys* bridges a morphological gap between forms with long and short carapaces. Carapace shape, however, is prone to homoplasy and shows no consistent relationship with trophic ecology, as demonstrated by new data, including a reappraisal of the poorly known *Pahvantia*. Despite distinct carapaces, *Titanokorys* shares similar rake-like appendages for sediment-sifting with *Cambroraster*, a smaller but much more abundant sympatric hurdiid from the Burgess Shale. The co-occurrence of these two species on the same bedding planes highlights potential competition for benthic resources and the high diversity of large predators sustained by Cambrian communities.

# 1. Introduction

Radiodonts are typified by an oral cone composed of multiple toothed plates, a pair of arthrodized frontal appendages, stalked

eyes and a trunk with lateral flaps bearing gills (e.g. [1–3]). The well-sclerotized appendages make up the majority of the radiodont fossil record and provide important characters that are informative about ecology and phylogeny. In hurdiids, the most diversified radiodont family, the frontal appendages are stout and rake-like with at least five elongate blade-like endites on the proximal podomeres, each bearing secondary spines or setae, specialized for sweep feeding on a variety of micro to macroscopic preys, suggesting that these animals occupied different niches [2,4,5]. This characteristic appendage distinguishes hurdiids from other radiodont groups, for example anomalocarids and amplectobeluids which evolved raptorial appendages, adapted for grasping of prey [6–9].

Comparatively less attention has been paid to other radiodont body parts such as their head sclerites. Although cephalic carapaces probably had deeper evolutionary roots among panarthropods (e.g. [10,11]) and are present in the form of small rounded sclerites in several non-hurdiid radiodont lineages including *Anomalocaris* [2,12] and *Amplectobelua* [13], many hurdiids stand out with respect to their oversized carapace complexes, consisting of a large central (H) and paired latero-ventral (P) elements [1,2,4,5,14].

An array of articulated specimens and probable moult assemblages, the majority described over the past decade, have considerably increased our knowledge of hurdiid anatomy and evolution and emphasized the diversity of carapace forms present within this group [2,4,5,14,15]. Several of these taxa, first described from the Burgess Shale, have now been recognized elsewhere [16–20]. This, in addition to new discoveries (e.g. [21]) and the reinterpretation of previously problematic fossils as hurdiids [22,23], suggests that this group of organisms had a broad geographical distribution, particularly during the Cambrian period [21].

Here, we present a new hurdiid discovered in the same Burgess Shale deposit and often on the same bedding surfaces as *Cambroraster falcatus* [2]. This rare form is distinctive in both the large size and shape of its carapace complex. This provides an impetus to apply geometric morphometric analyses, following pioneering morphometric work published on *Hurdia* from the Burgess Shale [4], with the aims of rigorously delimiting hurdiid species and examining allometry and the relationship between shape and ecology. The phylogenetic relationships of hurdiids have been based primarily on appendicular morphology [5,22,24] and are here reinvestigated in light of the new data provided in this study, using different approaches to incorporate carapace morphology. This is also assisted by a critical reinterpretation of fossil material belonging to the poorly known hurdiid *Pahvantia*. Finally, the co-occurrence of the new species with *Cambroraster* opens questions related to the ecological relationships among radiodonts within the Burgess Shale community.

# 2. Material and methods

## 2.1. Fossil material and preservation

This study is based on 12 Burgess Shale specimens of *Titanokorys gainesi* gen. et sp. nov. collected from Marble Canyon and Tokumm Creek in northern Kootenay National Park, British Columbia [25,26]. All specimens are housed at the Royal Ontario Museum (ROM), Invertebrate Palaeontology section (ROMIP; see electronic supplementary material, table 2.1). Specimens were photographed under direct or cross-polarized lighting conditions.

Only probable moult assemblages and isolated carapace elements are currently known (figures 1–4; see the reconstruction in figure 5). Most specimens are fragmentary, and several are deformed due to variations in the angle of burial (figure 3a–c). As is typical with Burgess Shale-type preservation, fossils are compressed and preserved as aluminosilicate and carbon films [28]. Fossils appear black on freshly exposed surfaces (figures 1 and 4). In some cases, they appear yellowish and have little contrast against the matrix (figures 2 and 3) due to weathering processes. Some specimens of *Titanokorys gainesi* gen. et sp. nov. (figure 2) are preserved on the same bedding plane as *Cambroraster falcatus*, a smaller hurdiid with a distinct horseshoe-shaped H-element [2], demonstrating that both species were buried together.

## 2.2. Morphometric analyses

Carapaces (H- and P-elements) of *Titanokorys gainesi* gen. et. sp. nov., *Cambroraster falcatus*, *Hurdia victoria*, *H. triangulata* and *Anomalocaris canadensis*, housed at the ROM, were photographed for shape analyses, and the shapes for other radiodont species were extracted from figured material in publications (electronic supplementary material, tables 2.2–3).

Carapace outlines were traced in Adobe Photoshop CS5. Shape analyses were performed in R v. 3.6.0 [29] using packages geomorph [30], Momocs [31], abind [32], ape [33], ggplot2 [34] and tntrtools [35]. Landmarks were placed at corresponding locations that were identifiable in all specimen outlines: for H-elements, one at the anterior and posterior and one at each tip of the pair of posterolateral processes (see electronic supplementary material for P-elements). A set of 10 semilandmarks was placed between each pair of landmarks along the outlines. In addition to landmark analyses, we also performed Elliptical Fourier Analysis based on the same specimen outlines, for comparison (see electronic supplementary material for details).

Landmark configurations were aligned using least-squares generalized procrustes analysis (GPA) (electronic supplementary material). Semilandmarks were slid to minimize bending energy. An additional procedure was performed to only retain the bilaterally symmetrical component of shape (electronic supplementary material).

Shape spaces were constructed with principal component analyses (PCA) on the Procrustes coordinates (figure 7). A preliminary analysis using a limited sample of P-elements revealed poor differentiation between species (see electronic supplementary material, figure S2), as previously reported within the genus *Hurdia* [4], and further analyses were not conducted with this data. We present the data for the H-elements below.

For *Cambroraster falcatus* and *Hurdia* spp., whose H-element sample sizes were sufficiently large, we used Procrustes ANOVA [2,36] to test for intraspecific ontogenetic allometry in H-element shape based on our landmark analysis. This method compares the fit of a linear (anisometric) model to a constant (isometric) null model of shape versus size (electronic supplementary material, figure S4). Based on the results of the fitted model, we estimated a series of landmark configurations corresponding to different stages in the ontogenetic series.

## 2.3. Phylogenetic analyses

We conducted parsimony analysis under equal and implied ($k = 3$ or 10) weights in TNT 1.5 [37], using a modified version of the matrix published in [2] incorporating a number of new characters dealing with carapace morphology (see electronic supplementary material for a list of modifications and electronic supplementary material, figures S5–7; see also [38] for results using a Bayesian approach).

We took two separate approaches to incorporate carapace shape information into our phylogenetic analysis. First, we constructed five discrete characters describing H-element shape, informed by our PCA results (D tree). We performed a second tree search directly incorporating the mean landmark configurations for each species (aligned by resistant fit theta rho analysis, RFTRA) into the analysis (L tree; see details in electronic supplementary material). Redundant discrete characters were excluded. For comparative purposes, we also conducted a third phylogenetic analysis excluding all H-element shape characters (N tree).

Finally, we plotted phylomorphospaces (figure 8; electronic supplementary material, figure S8) using a PCA on RFTRA-aligned mean H-element landmark configurations of all species which were sufficiently complete for inclusion and projecting the D and L trees into this space, with parsimony ancestral states reconstructed during tree optimization in TNT. For the D tree, this was accomplished by running an analysis optimizing the landmark character but constraining the topology to match that of the D tree. Ordinations were performed on RFTRA coordinates to remain consistent with the phylogenetic optimization procedure, but results using GPA (not presented) were qualitatively similar. Transitions in feeding ecological category were mapped onto the trees based on a parsimony criterion in Mesquite 3.40 [39].

## 2.4. Boxplots

Boxplots of the sagittal length and width at the tips of the posterolateral processes of the H-element for seven Cambrian hurdiid species were based on measurements of ROM specimens and published material (figure 9; electronic supplementary material, table 2.2).

# 3. Fossil descriptions

## 3.1. Systematic palaeontology for *Titanokorys gainesi*

Superphylum Panarthropoda Nielsen, 1995 [40]
Order Radiodonta Collins, 1996 [3]

Family Hurdiidae Lerosey-Aubril & Pates, 2018 [22]
*Titanokorys gainesi* gen. et sp. nov.
LSID urn:lsid:zoobank.org:act:F07D43CF-9422-4148-B8E9-AF26E0BC895D
LSID urn:lsid:zoobank.org:act:0EE750AC-FB83-4748-ADD9-013EB53459F1

### 3.1.1. Etymology

Genus name from *Titans*, a group of powerful Greek deities of great sizes, in reference to the large size of the central carapace element and from the Greek word *Korys* meaning helmet; *gainesi*, after Robert R. Gaines, Professor of Geology at Pomona College, who first joined the ROM-led field expeditions in 2008 as a research collaborator. Robert Gaines was instrumental in the co-discovery of the Marble Canyon fossil deposit in 2012 [26] and several related Burgess Shale outcrops along Tokumm Creek [25], including many fossils in this study.

### 3.1.2. Type material

Holotype—ROMIP 65415, a probable moult assemblage consisting of an H-(central) element, a pair of lateral (P) elements, a pair of frontal appendages, oral cone and probable gill blades (figure 3). Paratypes—ROMIP 65168, a complete H-element (figure 1); ROMIP 65741, an individual H-element with partial endites (figure 2); ROMIP 65748 and ROMIP 65749, two fragmentary H-elements (figure 4); other materials: seven additional H-elements, one associated with a P-element and poorly preserved appendages, in various states of preservation and completeness (electronic supplementary material, table 2.1).

### 3.1.3. Locality and stratigraphy

The upper part of the 'Thick' Stephen (Burgess Shale) Formation, Cambrian (Miaolingian Series, Wuliuan Stage), *Ehmaniella* biozone, from Marble Canyon [26,41] and Tokumm Creek [2,25] localities.

### 3.1.4. Diagnosis for genus and species

H (central)-element ovoid, bearing a broad anterior sagittal spine (*ca* 7% the length of the entire carapace), flanked by a pair of rounded processes pointing forward. Ocular notches moderately incised and positioned at the posterior of the H-element, flanking a short (*ca* 10–15% the length of the entire carapace), subtrapezoidal and slightly bilobate axial projection separated by a shallow medial notch. Spinous posterolateral processes are very short, about half the length of the bilobate axial projection. Elongate (sub-equal in length to H-element) and ovoid P (lateral)-element with a very short neck bearing a stout downward-pointing spine. Frontal appendages with elongate spiniform distal endites on podomeres 8–10 and elongate secondary spines on more proximal endites on podomeres 2–6; distal endites and secondary spines at least four times longer than the width of endite 6.

### 3.1.5. Description: paratype

ROMIP 65168 preserves a large (L: 27 cm; W: 18.5 cm) complete H-element in dorsal view and is the best example of this element for descriptive purposes (figure 1). The fact that it appears perfectly bilaterally symmetrical suggests that the carapace was buried and compacted with the dorsoventral axis perpendicular to bedding. The H-element is ovoid with the widest point about 60% from the front and width slightly decreasing along the posterolateral processes. Crescentic to linear compression artefacts, starting from the anterior along the midline and extending towards the lateral areas, suggests that the carapace was convex dorsally, curving down gently from the sagittal plane towards the margins. The front of the carapace yields a sagittal spine with a blunt tip (Sa in figure 1*a–c*). This spine is about 7% the total length of the carapace and is roughly the shape of an equilateral triangle. A pair of semicircular to somewhat angular bulges, or anterolateral processes, flank the sagittal spine (Ap in figure 1*a–c*). The base of each process is comparable in width to the base of the sagittal spine, but they protrude only a few millimetres forward.

The posterior margin of the H-element projects into a pair of posterolateral processes and a trapezoidal axial projection, separated by ocular notches (On in figures 1*a,b* and 3*a–c*). The axial projection is subtly bilobate with smooth rounded posterolateral margins separated by a shallow medial notch (Mn in figures 1*a,b* and 3*a–c*). It represents *ca* 10–15% the length of the carapace and

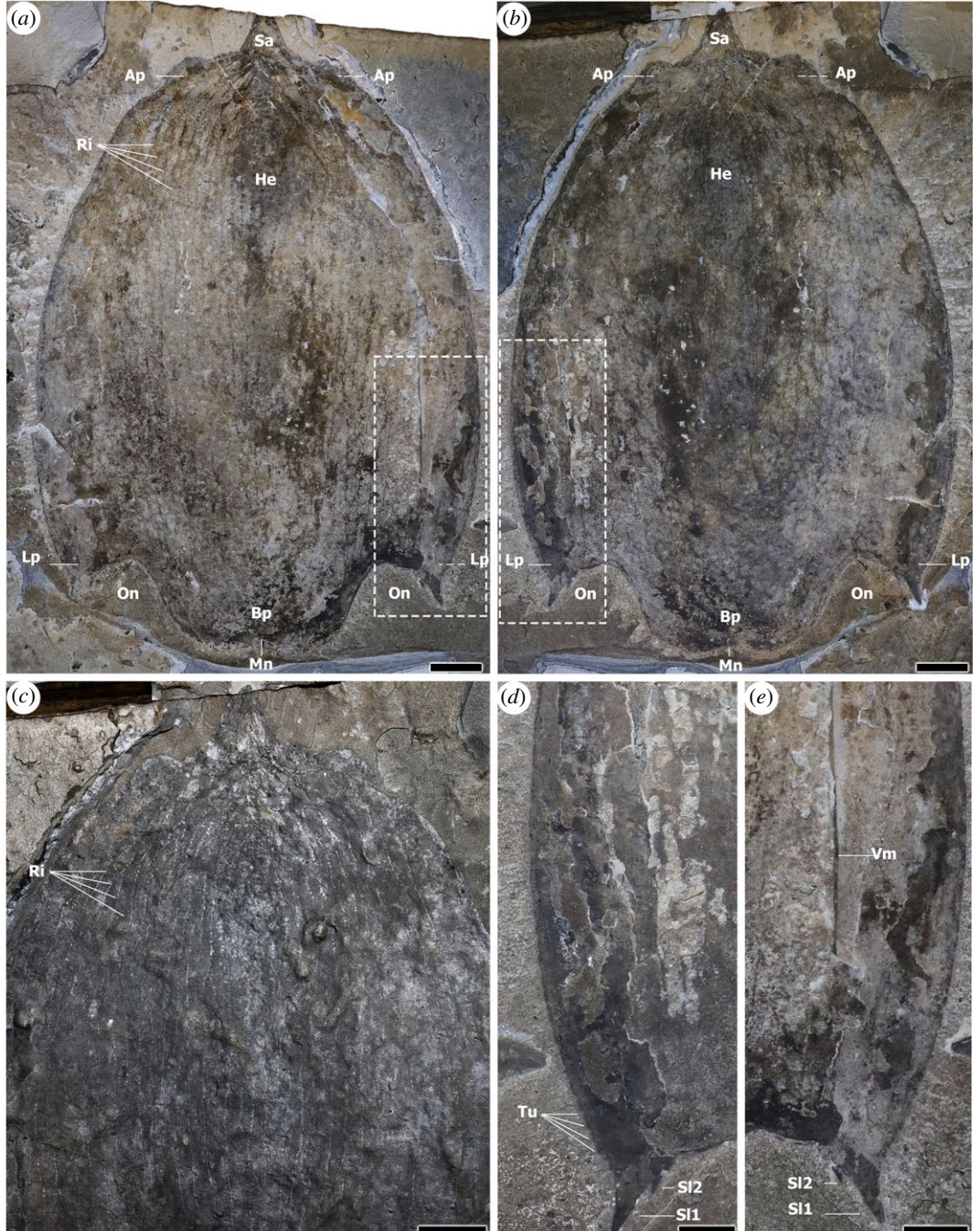

**Figure 1.** H-element of *Titanokorys gainesi* gen. et sp. nov., paratype ROMIP 65168. (*a*) Part; (*b*) counterpart; (*c*) close-up of ornamentation, photographed under low-angle light; (*d*), (*e*) close-ups of posterolateral margins. Ap, anterolateral processes; Bp bilobate axial posterior region; He, H-element; Lp, posterolateral processes; Mn, medial notch; On, ocular notch; Ri, ridges associated with a reticulated pattern; Sa, sagittal spine; Sl1,2, terminal (Sl1) and medial (Sl2) spines of posterolateral processes; Tu, tubercles; Vm, the ventrolateral margin of H-element. Scale bars: (*a*,*b*) = 20 mm; (*c*,*d*) = 5 mm.

60% of its maximal width anteriorly, becoming narrower towards the rear. This implies that the two ocular notches located on either side, are angled *ca* 20° anterolaterally from the midline. The posterolateral processes in *Titanokorys* are very short, extending only half as far posteriorly as the axial projection (Lp in figures 1*a*,*b* and 3*a*–*c*). Each process has a smooth ventral margin and terminates in a large spine (Sl1 in figure 1*d*,*e*). A second shorter spine, about 25% the dimensions of the large spine, is located along the laterodistal margin of the ocular notch, near the base of the large spine (Sl2 in figure 1*d*,*e*). Both spines are slightly curved such that they point posterolaterally.

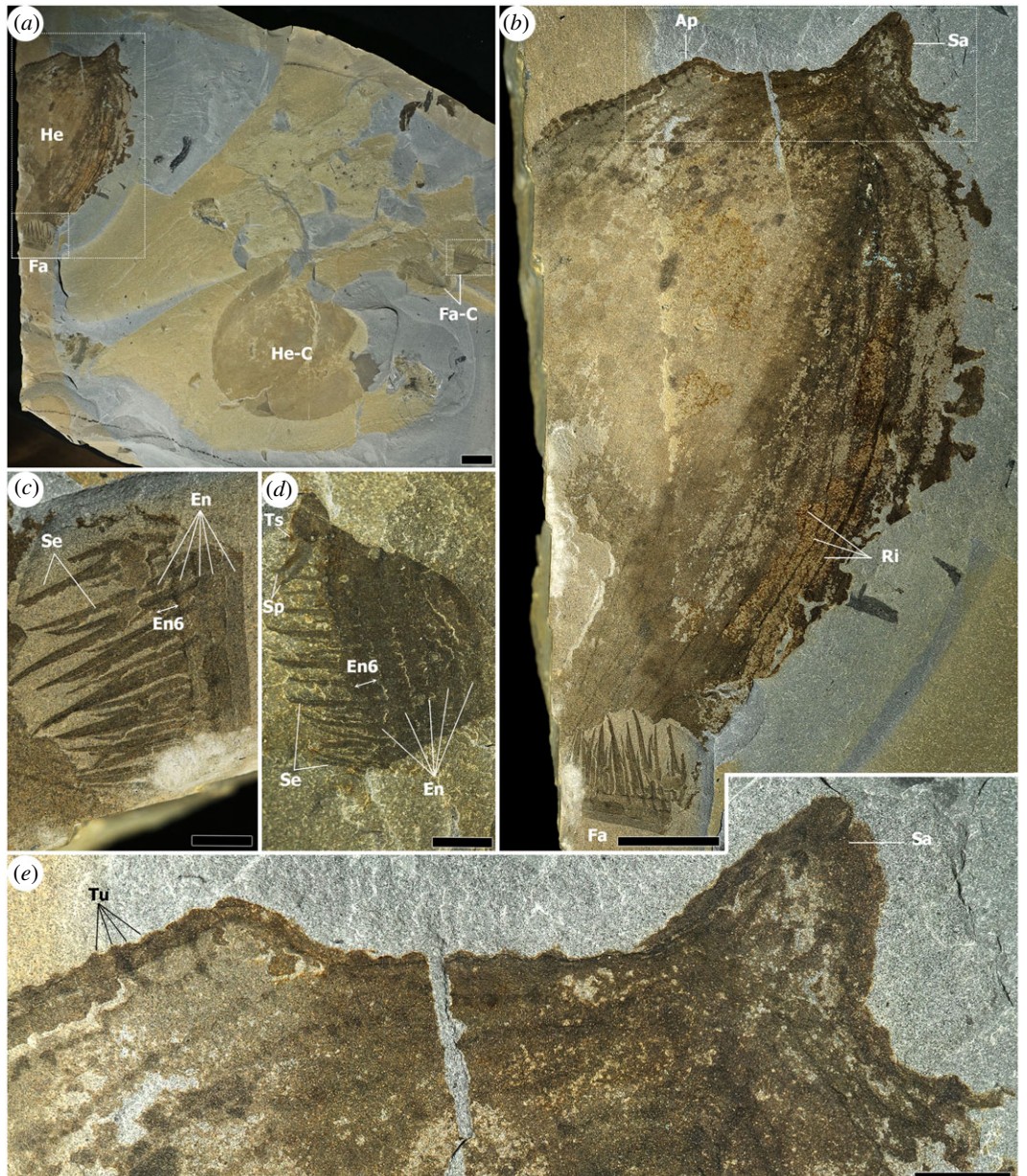

**Figure 2.** Assemblage of *Titanokorys gainesi* gen. et sp. nov., paratype ROMIP 65741. (*a*) Overview of slab, showing close association of H-element and partial appendage with an assemblage of *Cambroraster falcatus* consisting of an H-element and pair of appendages; (*b*) detail of *T. gainesi*; (*c*) close-up of endites from frontal appendage; (*d*) close-up of endites from frontal appendage of *C. falcatus*; (*e*) close-up of anterior margin of H-element, showing ornamentation. Ap, anterolateral processes; EnX, endite no. X; He, H-element; He-C, H-element of *C. falcatus*; Fa, frontal appendage; Fa-C, frontal appendage of *C. falcatus*; Ri, ridges associated with a reticulated pattern; Sa, sagittal spine; Se, secondary spines on endites; Sp, spiniform distal endites; Ts, Terminal spine; Tu, tubercles. Scale bars, (*a*,*b*) = 20 mm; (*c–e*) = 5 mm.

The entire H-element is ornamented by a series of longitudinal ridges running more-or-less parallel to each other (Ri in figures 1*a*,*c*, 2*b* and 4*a–c*). The ridges occasionally converge, resulting in a reticulated pattern, similar to *Cambroraster* [2] but with particularly elongate and angular cells. The number of ridges probably ranges between 30 and 40 in total across the entire width of the H-element as seen from the top. Each ridge consists of a reflective band dotted with chains of repeated structures, each about 1 mm wide and 2 mm long (figures 2*d* and 4*a–c*). Tubercles are present along the margins of the carapace, in particular anteriorly and along the outside margin of the lateral processes (Tu in figures 1*d*, 2*d*, 3*a* and 4*a–c*). By their comparable sizes and arrangements, these seem to represent the lateral expression of the dorsal repetitive spots suggesting that the carapace is covered by rows of

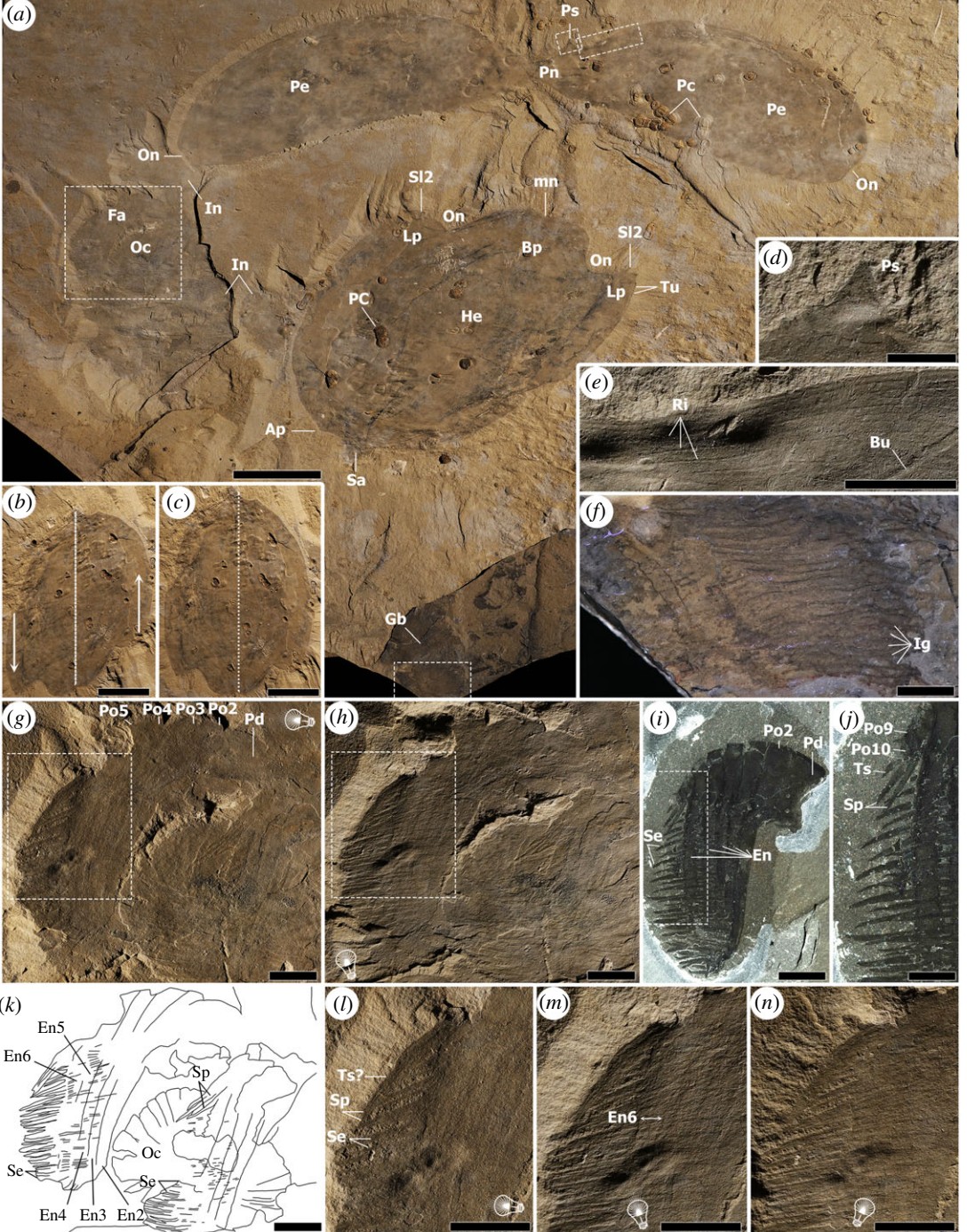

**Figure 3.** Assemblage of *Titanokorys gainesi* gen. et sp. nov., holotype ROMIP 65415. (*a*) Overview of slab, with boxed regions indicating close-ups in other panels, note associated agnostids (*Peronopsis* cf. *columbiensis*) possibly feeding on the remains or encrusting biofilms [27]; (*b,c*) original obliquely preserved H-element, with arrows showing the direction of deformation and dashes indicating sagittal axis of symmetry (*b*) and hypothetical undeformed version (*c*) using distort mode in Adobe Photoshop version 21.2.2 (based on the length-width proportions of ROMIP 65168). (*d*) Close-up of P-element spine; (*e*) close-up of P-element showing ridges; (*f*) close-up of bands of gill lamellae; (*g,h*) appendages and oral cone photographed using different low-angle light orientations to emphasize different details; (*i,j*) overall view (*i*) and close-up (*j*) of the frontal appendage of *Cambroraster falcatus*, ROMIP 65084, showing comparatively shorter spiniform distal endites and shorter secondary spines on more proximal endites. (*k*) Line drawing of appendages and oral cone of *T. gainesi* (from *g,h*); (*l–n*) close-ups of frontal appendages using different low-angle light orientations (*l*, close-up of *g*; *m*, close-up of *h*). Bu; burrow; Gb, gill blade; Ig, individual gill filament; In, Indeterminate; Oc, oral cone; Pc, *Peronopsis* cf. *columbiensis*; Pd, peduncle (podomere 1); Pe, P-element; PoX, podomere no. X; Ps, P-element spine; other abbreviations see figures 1 and 2. Scale bars: (*a–c*) = 50 mm; (*e,g–i,k–n*) = 10 mm; (*d,f,j*) = 5 mm.

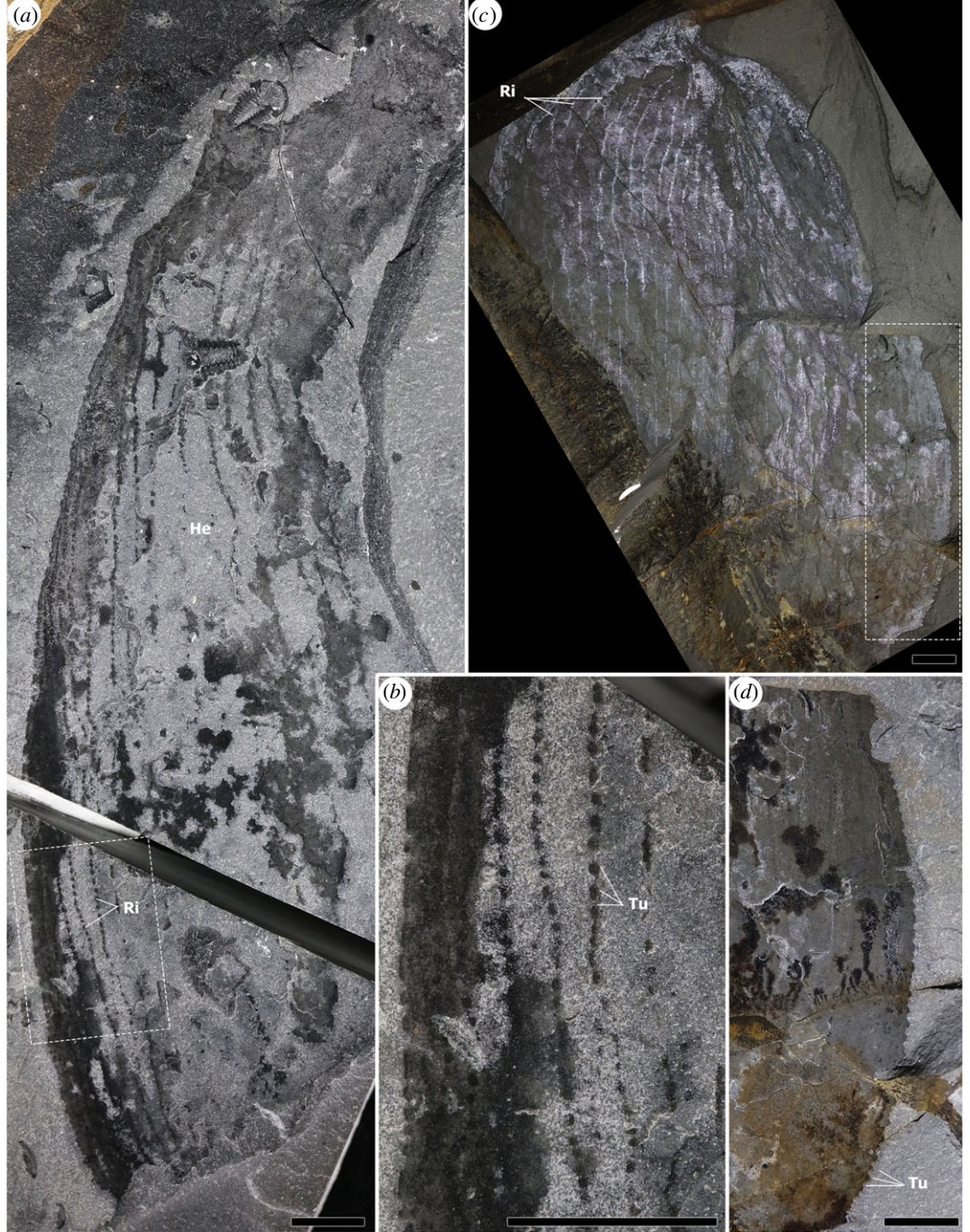

**Figure 4.** H-elements of *Titanokorys gainesi* gen. et sp. nov. showing ornamentation. (*a,b*) Paratype ROMIP 65749; (*a*) overview, note associated ptychopariid trilobites; (*b*) close-up of boxed region from (*a*); (*c,d*) paratype ROMIP 65748; (*c*) overview photographed under low-angle light; (*d*) close-up of boxed region in (*c*) showing tuberculate margin. For abbreviations, figure 1. Scale bars = 10 mm.

small elongated tubercles. These tubercles increase in size towards the terminal spine of the posterolateral processes (figures 1*d* and 3*a*).

The pair of ventrolateral (P-) elements are best preserved in the holotype ROMIP 65745 (figure 3*a*). The P-elements share the same overall dimensions (*ca* $L = 20$ cm and $H = 7$ cm) are almost perfect mirror images of each other and show only a few marginal compression artefacts suggesting they were buried more-or-less parallel to bedding. This is unlike the associated H-element, which by comparison with the paratype ROMIP 65168 (figure 1) is not bilaterally symmetrical, suggesting oblique burial. Retrodeformed longitudinally, the H-element, probably would have reached *ca* 21 cm

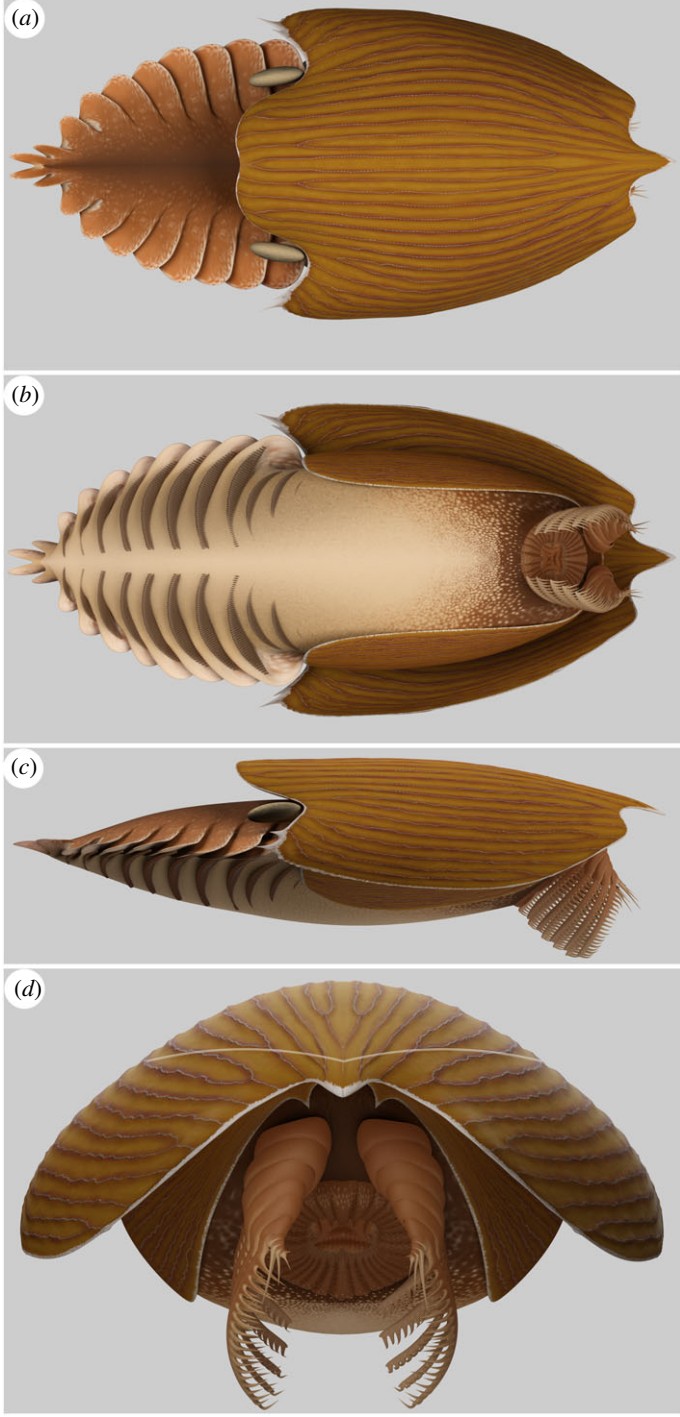

**Figure 5.** Reconstruction of *Titanokorys gainesi* gen. et sp. nov. (*a*) Dorsal view; (*b*) ventral view; (*c*) lateral view; (*d*) frontal view—white line represents the upper margin of the P-elements below the H-element. Reconstruction by Lars Fields (see electronic supplementary material, video file 1).

in length and *ca* 15 cm in width (assuming a similar L/W ratio as ROMIP 65168), which is subequal to the length of the P-elements (figure 3*c*). Each P-element is drop-shaped with a more linear dorsal and truncated posterior margins. The posterodorsal corners of the P-elements are slightly indented, probably representing ocular notches (On in figure 3*a*). The paired P-elements are connected to each other by a very short anterior neck section (Pn in figure 3*a*) which bears a single broad-based spine on its curved ventromedial surface (Ps in figure 3*a*,*d*). This spine is about a quarter the height of the neck above it. The carapace surface, as in the H-element, bears longitudinal rows of tubercles. In the

holotype (figure 3*e*), the rows are more visible along the ventromedial surface and appear thinner and with smaller tubercles compared with the H-element in ROMIP 65168 (figure 1). Based on previous studies [1,2,4], the P-elements would wrap tightly laterally and ventrally along the body, with their anterior attachment point below the base of the H-element sagittal spine (figure 5).

The holotype also includes a pair of frontal appendages and an oral cone (figure 3*b*) as well as a number of indeterminate elements. Smooth plates of the oral cone are faintly visible below or on top of other fossil material (Oc in figure 3*g,h,k*) and no fine details can be discerned. One appendage, in lateral view, can be distinguished more clearly using different low-angle light orientations (figure 3*g,h, k–n*). It preserves the peduncle (Pd) and all the podomeres (Po), with five long endites (numbered according to their respective podomeres, En2 to En6) bearing distally pointing, gradually tapering secondary (or auxiliary) spines (Se). About 35 secondary spines are visible beyond endite 6, but this number includes overlapping spines from more proximal endites as well, so the true number of spines per endite may be closer to 20–25. The secondary spines are also well preserved in ROMIP 65741 (figure 2*c*). Each of these spines on that specimen or the holotype is nearly four times longer than the width of endite 6, a third longer compared with equivalent spines in *C. falcatus* (figures 2*d* and 3*i,j*). It is unclear if the distal ends of the secondary spines were hooked like in *C. falcatus*, although they appear to be curved (figures 2*c* and 3*l–n*). The holotype shows evidence of unpaired spines and spiniform endites on more distal podomeres (probably 8, 9 and 10) (Sp in figure 3*g,h,k–n*). In addition to sclerotized elements, the holotype also preserves bands of lamellae, showing individual gill elements (Gb figure 3*a*; lg figure 3*f*). These are similar to those of other hurdiids (e.g. [1,5]) and were probably associated with the trunk and lateral flaps.

The reconstruction (figure 5) is based on all of the fossil material available. Details of the trunk, eyes and flaps are not preserved but have been hypothetically reconstructed based on *C. falcatus* [2].

## 3.2. Morphological reinterpretation of Pahvantia hastata

Lerosey-Aubril and Pates [22] recently described a fossil assemblage of *Pahvantia* (KUMIP 314089) showing a well-preserved tripartite carapace complex, identifying it as a radiodont, as well as what they interpreted as an unusual type of appendage with extremely long setose endites (their figure 3*a–c*). This appendage was interpreted as having seven endites divided into two broadly different types and sizes. The two proximal endites they identified were short with about seven strong auxiliary spines each, with the second endite being about three times wider compared with the first endite. The five distal endites were long—up to four times the size of the proximal endites—and had up to 50–60 setae.

There are a number of issues with the interpretation of the putative appendage in this specimen [2]. In particular, there is no evidence of podomeres associated with the elongate setose 'endites', and these 'endites' are irregularly bent, curving to varying degrees along their lengths, suggesting they were highly flexible. Endites are well sclerotized in other hurdiids and are therefore not typically deformed in this way [2,5]. Hurdiid endites generally exhibit a smooth mesial curvature [2], contrary to the sharp bends seen in the *Pahvantia* specimen. In addition, no hurdiid shows endites with clearly differentiated secondary spines and secondary setae in the same appendage. One possible exception is *Aegirocassis* which might have secondary spines on the peduncular endite, but these are not clearly preserved in the figured material (figure 2*a,b* in [5]). Finally, the presence of two rather than a single peduncular endite as well as their small relative size is unprecedented [42].

As is typical with Burgess Shale-type preservation, KUMIP 314089 is preserved as a part and a counterpart, with the split going through various superimposed layers of the fossilized tissues in such a way that different structures may be visible on each. The part is the best preserved; however, the counterpart shows some remnants of structures missing on the part (figure 6*b,c*). When images of both part and counterpart are superposed, the stacked image shows clearly the presence of a distinct appendage, *ca* 10 mm in height, partly overlapping a larger array of ribbon-like structures (figure 6*b*). The appendage bears five endites of similar lengths, each probably bearing seven to eight robust, hooked secondary spines and terminating distally in a similar hooked spine. These structures were partially described by Lerosey-Aubril and Pates [22] who interpreted them as two proximal endites, but we consider their 'second endite' to actually consist of three partially stacked endites with visible posterior margins. The most distal elongate endite (number six) was overlooked and is mostly preserved on the counterpart (only its tip is visible on the part) together with the dorsal sections of the podomeres, which are not preserved on the part. A semicircular structure proximal to the endite-bearing podomeres can be identified as the peduncle. Triangular projections dorsal and ventral to the peduncle might represent a dorsal spine and the remnants of a peduncular or shaft endite [42],

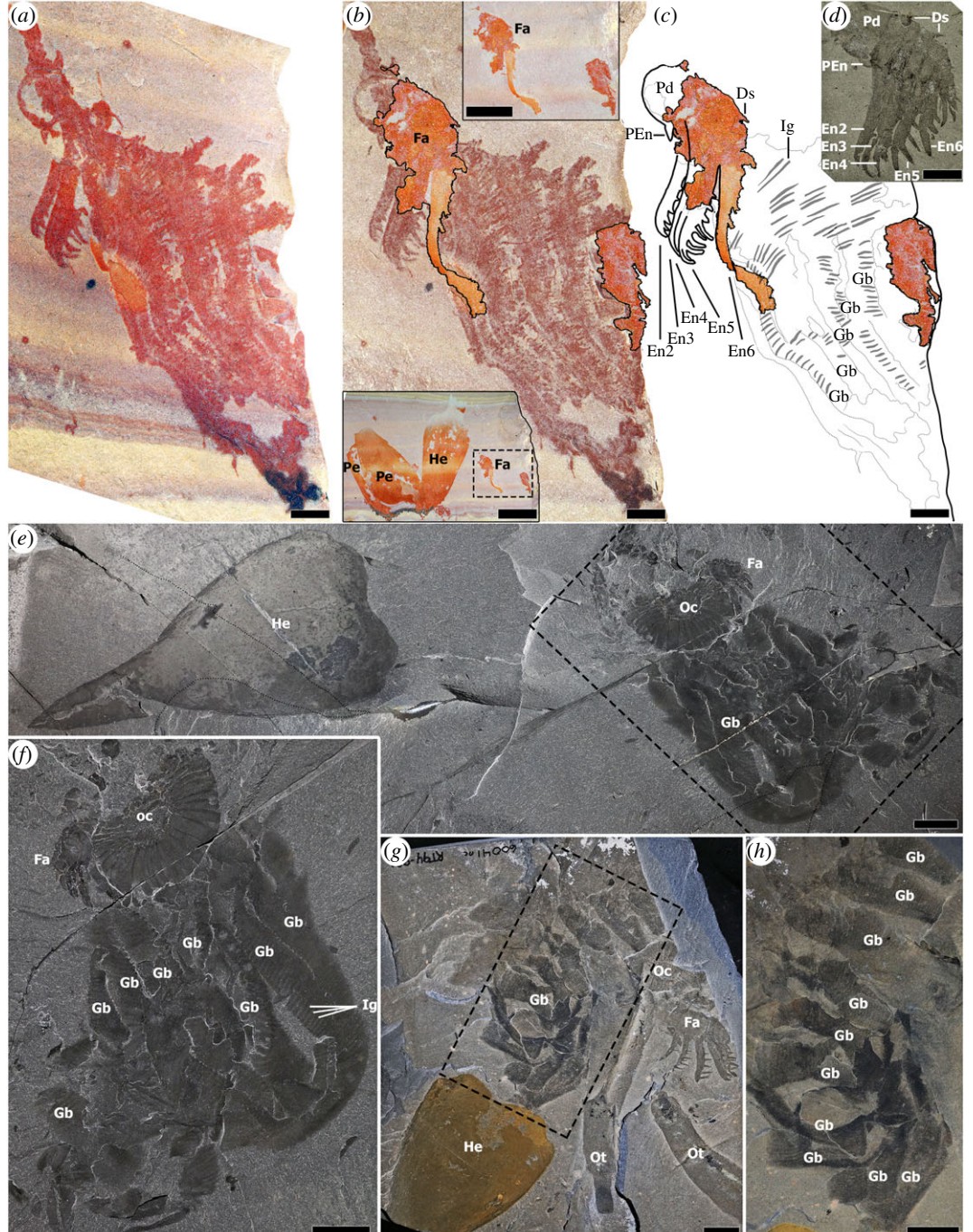

**Figure 6.** Comparative morphology of *Pahvantia hastata*. (*a–c*), *P. hastata* KUMIP 314089; (*a*), the part showing distal ends of broken endites to the left of the gill blades; (*b*) part and counterpart superposed to show the nearly complete appendage partly overlying the gill blades, lower inset showing complete counterpart with carapace elements, upper inset showing a close-up of partial appendage and gills on counterpart; (*c*) counterpart superposed on line drawing of part; (*d*) appendage of *Hurdia* for comparison, ROMIP 59259; (*e–h*), disarticulated *Hurdia* assemblages, showing groups of connected gill blades associated with other body parts; (*e,f*), ROMIP 60031; (*g,h*), ROMIP 60041. Scale bars: (*a–c*) = 2 mm; (*b*); upper inset, 5 mm; lower inset, 10 mm; (*d–h*) = 10 mm. Ds, dorsal spine; Ot, *Ottoia prolifica*; PEn, peduncular endite, other abbreviations see figures 1 and 3. (*a–c*) Images courtesy Rudy Lerosey-Aubril.

respectively. Additional possible dorsal spines are present more distally. Overall, the newly recognized appendage is about half the maximum dimension of the structure previously described by Lerosey-Aubril and Pates [22]. The appendage, from the distal tips of its endites to the outer margin of the podomeres, also represents less than one-third of the length of the associated H-element on the same

slab, which would make it more consistent with the carapace/appendage size ratio observed in other hurdiid assemblages such as *Hurdia* [4] and *Cambroraster* [2].

The putative setose endites described by Lerosey-Aubril and Pates [22] are more comparable to structures interpreted as gill blades, associated with flaps or body segments in anomalocaridids and hurdiids [4,12,15] including *Hurdia* (figure 6*e–h*). These bands of lamellate structures are not part of an appendage, nor are they likely to have been used for feeding (contra [22]), probably functioning instead for respiration [15]. In *Hurdia*, the gill blades are prominent ventrolaterally and can be very elongate as demonstrated in disarticulated or isolated material [4]. Stacks of *Hurdia* gill blades can be preserved in similar ways to those observed in the *Pahvantia* material, especially in disarticulated carcasses or moult assemblages (figure 6). In addition, the number of individual elements in the bands in *Pahvantia* is roughly similar to what is known in the gills of *Hurdia* [4] suggesting they are equivalent structures. The amorphous strand of material overlapping and extending beyond the margin of the appendage peduncle, originally interpreted as a part of the appendage, is more likely associated with the mass of gills and trunk cuticle.

In conclusion, a re-evaluation of KUMIP 314089 based on the photographic material provided by the authors, leads us to demonstrate the presence of a nearly complete and partially unnoticed appendage. This appendage has the greatest similarity in morphology, particularly in terms of the number and form of the secondary spines, to *Hurdia* [2,4], suggesting an adaptation for capturing larger prey living along or in the sediment and thus probably a nektobenthic lifestyle, contra [22]. However, the subequal length of the five elongate endites as well as the shape of the H- and P-elements sets *Pahvantia* apart from *Hurdia*. Together with our phylogenetic results (see below), these differences justify the retention of *Pahvantia* as a distinct genus.

# 4. Results

## 4.1. Carapace shape variation and allometry

Taxa in our H-element shape space (figure 7; electronic supplementary material, figure S1)—including the most well-sampled taxa (*Hurdia*, *Cambroraster*, *Pahvantia*)—are broadly distributed and generally non-overlapping, with species identification accounting for 85% of the variance in the data (see electronic supplementary material). This suggests that discrete characters used for taxonomic purposes provide a good overall qualitative estimate of carapace variation. The first axis (PC1), which represents the largest amount of information (69% of variance), generally corresponds to the sclerite length–width ratio, largely driven by variations in the relative length of the anterior section of the carapace. At the left extreme of this axis, the anterior section of the carapace tends to be shortened resulting in a margin that is rounded or horseshoe-shaped (*Cambroraster*, *Zhenghecaris*) or broad and converging towards a very short spine (*Hurdia triangulata*). The right extreme of the first axis includes forms with very elongate anterior sections usually terminating in a sharp angle (*Pahvantia*, *Aegirocassis*) or spine (*Hurdia victoria*). *Titanokorys* occupies a distinct region of morphospace intermediate between *Pahvantia* and *Cambroraster*, with all of these forms separated from *Hurdia* along axis two (PC2). Some variation in anterior pointedness is also captured by axis two, with the most extreme case, represented by *Hurdia victoria*, occupying a high position on both of the first two axes. However, the second axis (14% of variance) relates primarily to the elongation of the axial posterior region of the carapace. Taxa lower on the second axis—*Titanokorys*, *Pahvantia*, *Cordaticaris* and *Cambroraster*—have well-developed posterior axial projections. High values represent forms with a shorter posterior axis such as *Hurdia* spp. and most extremely *Zhenghecaris*. The third axis (6% of variance) mainly separates *Aegirocassis* and to a lesser extent *Zhenghecaris* from all the other hurdiid taxa. In these forms, the posterolateral processes have become recurved anterolaterally leading to a relative expansion of the posterior region of the carapace. *Cordaticaris*, which is one of the most average taxa on axes one and two, plots towards the opposite end of axis three compared with *Aegirocassis*, consistent with a medially contracted posterior margin.

Species clusters of well-sampled taxa can occupy a large amount of morphospace suggesting intraspecific and taphonomic variation. Both putative *Cambroraster* specimens recently described from China [17,18] plot near the *Cambroraster* cluster (although at opposite extremes) supporting their taxonomic assignment to this genus. *Hurdia* cf. *victoria* [16] from Utah similarly plots at the edge of the *H. victoria* cluster.

We tested *Hurdia victoria* ($n = 35$), *H. triangulata* ($n = 20$) and *Cambroraster falcatus* ($n = 34$) for ontogenetic allometry (other taxa lacked sufficient sample sizes). Only *Cambroraster* showed significant

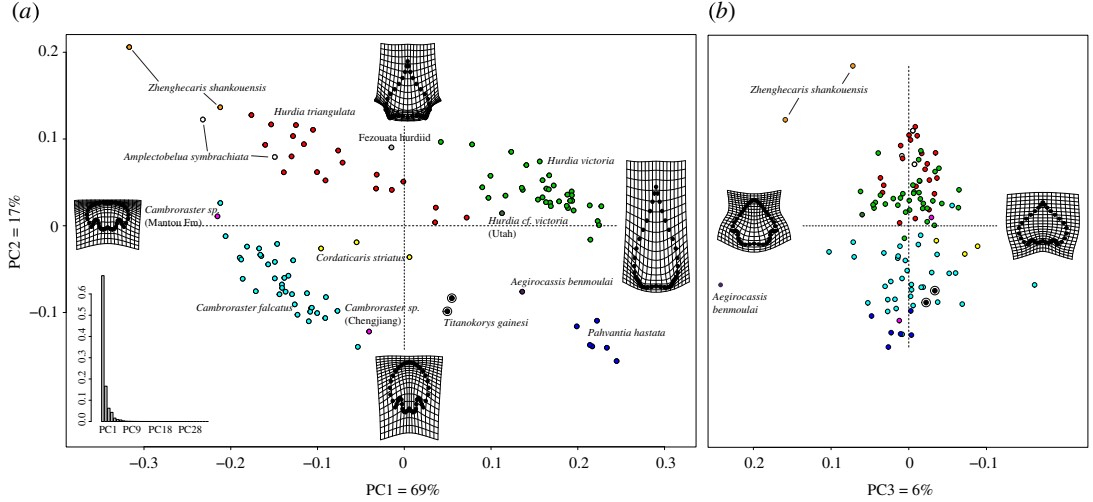

**Figure 7.** PCA of radiodont H-element shape based on landmark analysis. (*a*) Axes 1–2; (*b*) axes 2–3. Different species and morphs colour coded, deformation grids representing average shapes at the extrema of respective dashed axes, bar plot at the bottom left showing the per cent of variation explained by each axis.

anisometry ($R^2 = 0.20$, $F = 7.85$, $p = 0.002$; see electronic supplementary material for further details). As *Cambroraster* gets larger, the axial posterior area tends to become broader and the posterolateral processes become shorter and reflexed anterolaterally (electronic supplementary material, figure S4). Extrapolating this model beyond the upper size limit of known specimens results in a 'hypermorphic' form in which the axial area extends well beyond the outwardly angled posterolateral processes. Extending the prediction in the other direction results in a putative juvenile form with a strongly horseshoe-shaped H-element with inwardly directed posterolateral processes extending considerably beyond the narrow axial area.

The structure of the morphospace of species-mean shapes (used in the phylomorphospaces) is similar to that including all specimens, although the variation previously encompassed by axes two and three in the latter is somewhat compressed into axis two in the former (figure 8*b*). This results in greater emphasis on the distinctiveness of *Aegirocassis* which was represented only by a single well-preserved specimen.

## 4.2. Phylogeny

The addition of new species and modifications to the character matrix, in particular our reinterpretation of *Pahvantia*, led to increased resolution of the interrelationships of hurdiids relative to previous analyses [2] (see also [38]). Characters supporting each clade can be found in the electronic supplementary material (table 1.12).

All analyses (figure 8*a*, electronic supplementary material, figures S5–7) consistently find a clade composed of *Titanokorys*, *Cambroraster*, *Zhenghecaris* and *Cordaticaris*, although this clade receives low resampling support. *Pahvantia*, *Hurdia* and *Aegirocassis*, respectively, are successively nested sister taxa to this clade. The clade of all aforementioned taxa with large carapaces is found in a polytomy with *Stanleycaris* and *Peytoia*. *Schinderhannes* is sister to all other hurdiids included in our analysis. These results are robust to different weighting regimes. Non-hurdiids are largely unresolved under equal weights, as found previously [2]. Under implied weights resolution of the strict consensus trees is improved with amplectobeluid (*Amplectobelua* + *Lyrarapax*) and tamisiocaridid (*Tamisiocaris*+'*A.*' *briggsi*) clades recovered consistently and more variable and poorly supported relationships among other taxa. This serves to emphasize the relatively high homoplasy and character lability in these taxa.

The inclusion of carapace shape characters variably leads to increased or decreased resolution and branch support, especially for taxa known from few characters, indicating carapace shape is not always congruent with characters derived from other body parts. The choice of the method also results in some differences. In the D tree, we find a clade of *Cambroraster* and *Zhenghecaris*, together with *Titanokorys* and *Cordaticaris*. In the L and N trees, *Titanokorys*, *Cordaticaris* and *Zhenghecaris* form

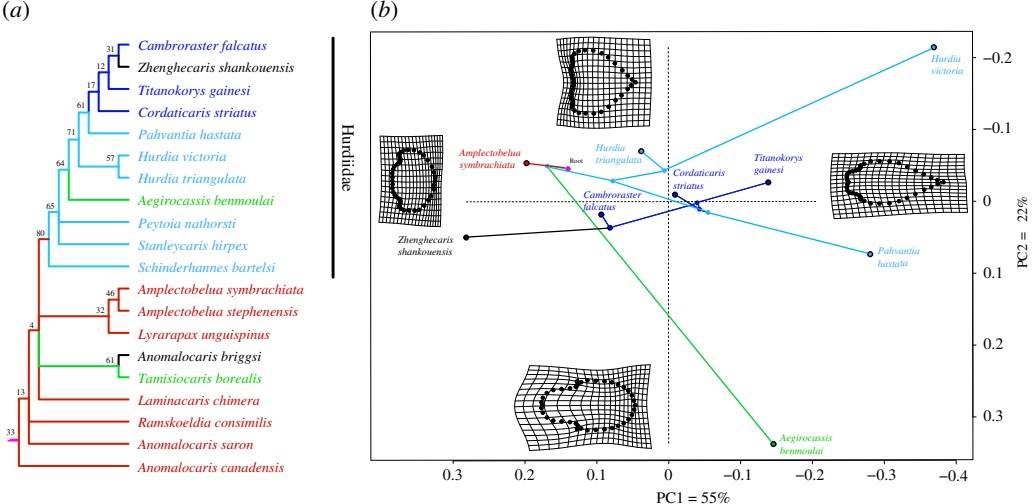

**Figure 8.** Evolution of radiodont H-element shape and feeding ecology. (*a*) Radiodont clade cut from a parsimony strict consensus tree optimized under implied weights (k = 3) with discrete characters describing carapace shape (D tree), colours (unrelated to those in figure 7) represent inferred feeding ecologies with parsimony ancestral states mapped over branches (red, macrophagous raptorial predator; green, suspension feeder; light blue, macrophagous sediment sifter; dark blue, microphagous sediment sifter; black, unknown; purple dot, root), numbers at nodes are symmetric resampling supports; (*b*) PCA phylomorphospace based on the same topology (excluding taxa for which shape is incompletely known), plotting RFTRA-aligned mean H-element shape for each species, with ancestral states estimated in TNT by optimizing the landmark configurations on a constrained topology, colours as in (*a*). (See electronic supplementary material for complete tree topology and results using alternative methods).

a clade sister to *Cambroraster*. *Titanokorys* and *Cordaticaris* are resolved as sister taxa in the L tree. The L and N trees are generally more similar suggesting that landmark data has little impact on the topology. This could be a result of the expected underweighting of landmark data relative to discrete characters because of the treatment of the entire configuration as one character [37]; however, even increasing the weight of the landmark character to be equivalent to the five overlapping discrete characters (assuming equal weights) results in no change to the topology (unpublished data). Discrete characters also have the advantage of permitting the inclusion of information from taxa with incompletely known carapaces, like *Peytoia* and *Lyrarapax*, albeit at a cost of precision in the description of shape.

Based on our phylomorphospaces, carapace shape appears to be prone to homoplasy, with closely related species often widely dispersed and overlapping regions occupied by more distant relatives (figure 8*b*; electronic supplementary material, figure S8). By contrast, mapping feeding ecology over the tree (figure 8; electronic supplementary material, figure S8) shows that it is strongly phylogenetically distributed, with feeding modes often evolving only once and being conserved in clades. This result is of course strongly influenced by the character richness of radiodont frontal appendages relative to other body parts.

# 5. Discussion

## 5.1. Comparative morphology

*Titanokorys* co-occurs with *Cambroraster* and both are even preserved on the same bedding surfaces (figure 2), providing strong evidence of sympatry. The two also possess some obvious similarities in morphology, such as the broad rounded H-element with wing-like posterolateral projections and frontal appendages with long pectinate endites bearing numerous closely spaced secondary spines. These observations might be considered evidence that the two represent dimorphic members of a single species; however, other evidence, detailed below, speaks against this notion.

*Titanokorys* possesses a mixture of characters seen in other radiodont taxa. The carapace ornament is essentially identical to the condition in *Cordaticaris* [17] and to a lesser extent *Zhenghecaris* [23], with the reticulate pattern being shared more broadly among many hurdiids, including *Hurdia* [4] and probably representing an external ornamentation similar to *Tuzoia* [43]. The ocular notches, and especially the medial notch, in *Titanokorys*, are less deeply incised than those in *Cambroraster*. The single

posteromedially directed spine in each ocular notch and lack of spines along the lateral margins of the H-element are also more reminiscent of *Zhenghecaris* than *Cambroraster*. While the H-element of *Titanokorys* is broadly rounded anteriorly, the anterior sagittal spine is more comparable to *Hurdia* and sets it apart from *Cambroraster* and *Zhenghecaris*. The small anterolateral processes in *Titanokorys* are probably homologous with previously unrecognized anterolateral processes present in *Hurdia* ([1] their figure 2*a* and [4] their figures 9*a*,*e*; 17*c*,*d*). In *Hurdia*, these structures are only visible on strongly tilted specimens, suggesting that they had a more ventrolateral expression in this taxon and could potentially be present and as yet unobserved in other hurdiids. Owing to the frontal position of these anterolateral processes and the relative lengths of carapace elements, the processes might be associated with zones of attachment of the P-elements. This is consistent with what is known about the association of H- and P-elements in *Hurdia*.

A chimeric mixture of characters is also seen in the P-elements of *Titanokorys*. The small posterodorsal notches appear to correspond to the ocular notches in *Hurdia* [1,4]. By contrast, these structures are notably absent in the more lenticular P-elements of *Cambroraster*. The ventrally directed spines emerging from the P-element necks are unique to *Titanokorys*. Irregularly rounded foramina observed in the P-elements of *Cordaticaris* were interpreted as possible muscle scars, or less probably, the bases of spines ([21] their figure 4*a*,*d*,*f*,*g*), and similar structures also occur in *Cambroraster* ([2] their figure 1*l*, *m*). Even if the latter interpretation is correct, these spines would be distinct from those in *Titanokorys* due to their projection perpendicular to rather than parallel to the plane of the carapace. In addition, the P-elements in *Cambroraster* are much shorter compared with the length of the H-element, and their neck areas are thin and elongate. In *Titanokorys*, the P-elements are almost as long as the H-element and the neck area is very short and wide.

Details of the other body elements are more difficult to compare owing to the lower fidelity of preservation. We are unable to discern whether the secondary spines on the appendages in *Titanokorys* are terminally hooked, as in *Cambroraster* [2]; however, the distal spiniform endites and the secondary spines are very elongate and their lengths represent at least four times the width of endites 2–6. In *Cambroraster*, the spines are shorter, *ca* twice to three times the width of endites 2–6. While subtle, this character tentatively differentiates the otherwise similar frontal appendages of these two species.

The distribution of H- and P-element shapes in *Titanokorys* does not overlap with those of *Cambroraster*. The largest *Cambroraster* specimen exceeds the width (though not length) of the smallest *Titanokorys* from which a reliable measurement can be obtained (figure 9). Additionally, the H-element of *Titanokorys* does not fall along the predicted allometric trajectory of *Cambroraster* (electronic supplementary material, figure S4). Interestingly, the modelled 'hypermorphic' *Cambroraster* instead resembles forms with extreme posterolateral expansion, particularly *Zhenghecaris* and to a lesser extent *Aegirocassis*, suggesting that heterochrony could have played a role in these divergences in shape. More detailed consideration is hindered by the statistical insignificance of allometric relationships in *Hurdia* spp., possibly an artefact of taphonomic noise and modest sample sizes, as well as the paucity of material for other radiodont species.

The above-noted differences in morphology between *Titanokorys* and *Cambroraster,* alongside the fact that the two are never found as sister taxa in our phylogenies (figure 8*a*; electronic supplementary material, figures S5–7), argue that these forms represent distinct sympatric taxa rather than ontogimorphs or sexual dimorphs in a single variable species. The delimitation of *Titanokorys* as a new genus is further supported by the recognition of potentially new species of *Cambroraster* [17,18] which are much more morphologically similar to *C. falcatus* than *T. gainesi*, as shown in our morphospace.

## 5.2. Ecological and evolutionary implications

The tripartite head carapace complex of *Titanokorys* reaches substantially larger sizes than the largest specimens from other Cambrian hurdiids (figure 9). Considering that in *Cambroraster* or *Aegirocassis*, the central (H-) element represents about half the total body length, by extrapolation *Titanokorys* could have reached 50 cm or more assuming comparable body/carapace length ratios between species. This makes *Titanokorys* one of the largest known hurdiids, and the largest Cambrian form in terms of its carapace complex. It is also one of the largest radiodonts on record from the Cambrian, only definitely topped by *Anomalocaris* which has been estimated to reach *ca* 60 cm [22] to 1 m [6]. Such early large body size foreshadowed the gigantism achieved by post-Cambrian hurdiids [5,14].

Given the highly distinct carapace morphologies in *Titanokorys* and *Cambroraster,* it is tempting to posit that carapace shape may have been related to ecological differences that facilitated niche

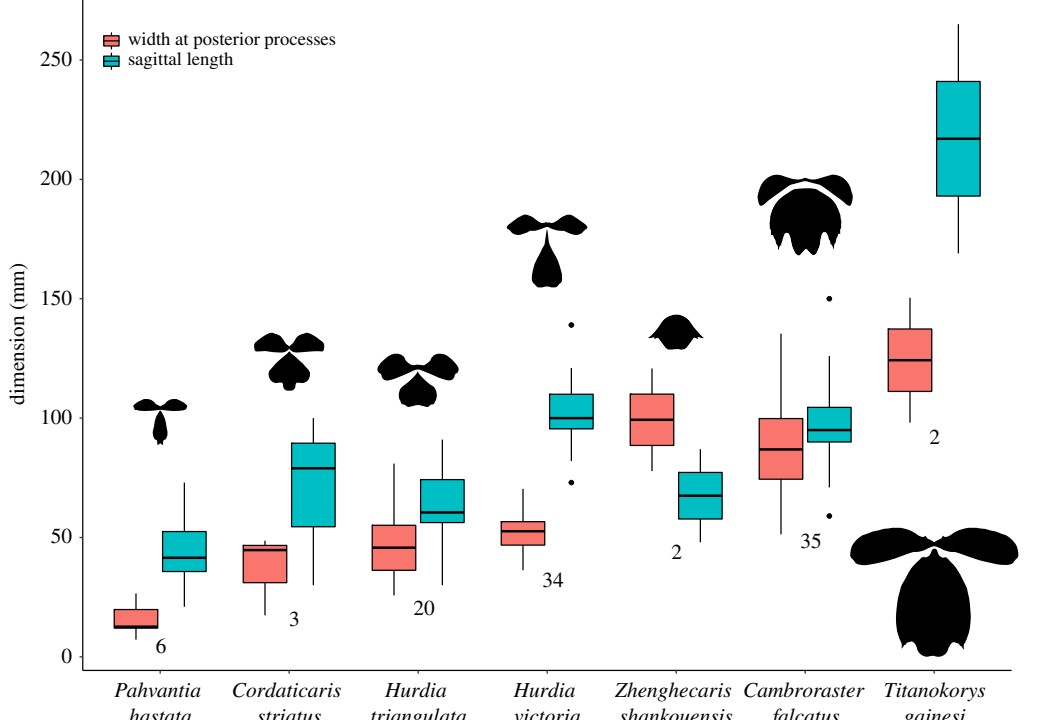

**Figure 9.** Boxplot of H-element sagittal lengths and widths (measured between the tips of the posterolateral processes) for Cambrian hurdiids. The numbers below boxes indicate number of sampled specimens. Silhouettes scaled to equal maximum relative size.

partitioning in these sympatric and otherwise similar species. A general association of hurdiid carapace shape and ecology has been hypothesized, with broad forms considered to correlate with benthic habits and more streamlined elongate forms like *Pahvantia* and *Aegirocassis* viewed as pelagic suspension feeders [18]. Our results do not favour this interpretation.

To begin with, we regard *Pahvantia* to possess similar appendages to *Hurdia*, with strong hook-like secondary spines (see above), and it is therefore not a suspension feeder, contra [22]. The only hurdiid providing compelling evidence for suspension-feeding based on appendage morphology is *Aegirocassis* [5,14]. *Aegirocassis* is strongly separated from other taxa in H-element shape space, although predominantly by its extreme posterolateral expansion rather than overall elongation (figures 7 and 8*b*). Further, this differentiation could either be related to differing ecomorphological adaptation or a consequence of chance coupling of shape and feeding ecology during phylogenetic divergence. Unfortunately, disentangling these two possibilities statistically is not possible at this time since so few species are known thoroughly.

Further insight can be gained qualitatively from considering the distribution of more subtle ecomorphological differences in phylomorphospace. On one hand, *Titanokorys*, *Cambroraster* and *Cordaticaris* have large numbers of long, rigid secondary spines (at least 20 per endite) which were interpreted as adaptations for a demersal, primarily microphagous sediment sifting lifestyle by comparison with extant euarthropods (e.g. [2]). *Hurdia* spp. and *Pahvantia*, on the other hand, have fewer (less than 10 per endite), more robust secondary spines, with larger interspace between them, presumably adapted to capturing larger prey. The bodies of *Hurdia* spp. are also more fusiform compared with *Cambroraster*. Considering H-element shape within these two groups, it is evident that closely related forms with similar feeding ecologies can be quite disparate, exemplified by the gap between *Hurdia victoria* and *H. triangulata*. Conversely, there is also overlap between the *Hurdia*–*Pahvantia* and *Cambroraster*–*Titanokorys*–*Cordaticaris* clusters in phylomorphospace, with *Titanokorys* notably extending the microphagous cluster into the region otherwise occupied by macrophagous taxa. Together these observations would seem to speak against a simple, direct relationship between carapace shape and prey size niche or associated habitat. However, the absence of an obvious mapping between carapace shape and trophic ecology need not imply that the hurdiid carapace was ecologically irrelevant. By comparison, the carapace in extant crustaceans can be variably adapted for

defence, visual communication or hydrodynamic performance as well as serving roles in feeding, with shape variation reflecting the complexity of these multiple roles [44].

The clade of the aforementioned hurdiid taxa, including *Titanokorys* and *Aegirocassis*, are characterized by large, extensively projecting carapaces, short bodies and stout appendages. Our phylogenetic results indicate that these morphologies are derived, having emerged from within a grade of hurdiids with smaller carapace complexes, including *Schinderhannes* [45], *Peytoia* [8] and probably also less well-known taxa like *Stanleycaris* [38,46] and *Ursulinacaris* [42]. Members of this grade also retain some plesiomorphic appendicular conditions, like a more elongate distal region with numerous robust podomeres, resembling the raptorial appendages of anomalocaridids and amplectobeluids [2]. The set of (usually) five elongate endites appears to have arisen in the hurdiid ancestor, at first with few, robust secondary spines, becoming increasingly numerous, thin and finely spaced in putatively pelagic (*Aegirocassis*) or derived demersal forms with enlarged carapace complexes [20] including *Titanokorys*. While changes in carapace shape within this derived lineage cannot be readily linked to changes in feeding ecology, it is possible that the initial enlargement of the carapaces constituted an adaptation to an increasingly specialized, typically benthic niche, serving for example for sediment ploughing or as a filtration chamber [18]. Alternatively, this enlargement could be a side consequence of other factors, such as the reduced requirement for agile motility in specialized benthic forms which could have enabled the evolution of bulky defensive armament.

Given the lack of evidence for a general correlation between carapace shape and feeding ecology, differences in body size between *Titanokorys* and *Cambroraster* might instead have played a part in selective resource exploitation, with *Titanokorys* presumably consuming larger prey. Intra-guild competition among these hurdiid predators is another possibility. Variation in spatial distribution might then have played a role in coexistence, with the relative rarity of *Titanokorys* in the Marble Canyon area possibly resulting from proximity to the edge of its range. Regardless of the exact ecological interactions between these species, this study strengthens recognition of the Cambrian benthos as a rich habitat for an array of large predatory animals to exploit [2].

Data accessibility. All supporting material can be found in the associated electronic supplementary material [47].

Authors' contributions. J.-B.C. led the ROM Burgess Shale expeditions which resulted in the collection of the fossil material published in this paper. J.-B.C. wrote an initial draft of the manuscript, prepared and photographed the material and assembled the fossil image figures. J.M. participated in fieldwork, compiled the morphometric and phylogenetic datasets, ran the analyses and created the associated figures. Both authors designed the study, discussed and interpreted the results and were involved in writing the final version of this manuscript.

Competing interests. We declare we have no competing interests.

Funding. Major funding support for fieldwork comes from the Royal Ontario Museum (Research and collection grants, Natural History fieldwork grants), the Polk Milstein Family, the National Geographic Society (no. 9475-14 to J.-B.C.), the Swedish Research Council (to Michael Streng), the National Science Foundation (NSF-EAR-1556226, 1554897) and Pomona College (to Robert R. Gaines). This research was also indirectly supported by the Dorothy Strelsin Foundation (ROM). J.M.'s research is supported by a National Science and Engineering Research Council (NSERC) Vanier Canada Graduate Scholarship through the University of Toronto (Dept. of Ecology and Evolution) and J.-B.C.'s NSERC Discovery grant (no. 341944). This is the Royal Ontario Museum Burgess Shale project no. 88.

Acknowledgements. Fossils were collected by Royal Ontario Museum field parties under several Parks Canada Research and Collections permits to J.-B.C. (KOONIP 2014-16317; YNP-2016-21639 and KOONIP 2018-2817). We thank T. Keith from Parks Canada for facilitating fieldwork activities and the 2014, 2016 and 2018 field crews. We thank R. Lerosey-Aubril for images of *Pahvantia* (figure 6a–c). L. Fields produced the artistic reconstructions (figure 5). M. Akrami and P. Fenton are thanked for assistance in the collections. J.M. would like to thank M. Collyer for troubleshooting assistance with geomorph and K. Nanglu, C. Aria, A. Izquierdo López and J. Moon for helpful discussions. This work was assisted by a geometric morphometrics workshop led by M. Silcox's lab at the ROM. TNT is freely available thanks to the Willi Hennig Society. We thank two anonymous reviewers for constructive feedback.

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
