## [Peer Review File · Royal Society Open Science]

Review History

RSOS-202138.R0 (Original submission)

Review form: Reviewer 1

Is the manuscript scientifically sound in its present form?

Yes

Are the interpretations and conclusions justified by the results?

Yes

Is the language acceptable?

Yes

Do you have any ethical concerns with this paper?

No

Have you any concerns about statistical analyses in this paper?

No

Recommendation?

Accept with minor revision (please list in comments)

Comments to the Author(s)

p. 2, l.21 The Abstract refers to 'sediment-sifting' yet the first paragraph describes hurdiids as 'specialized for sweep feeding ... in the water column'?

l.22 'zeppelin-shaped' is used in the Abstract and nowhere else (i.e. not in the Description). The outline of the carapace does not resemble a zeppelin.

l. 26 'a complex and varied relationship with ecology' is no doubt true, but it's also a euphemism for a lack of results.

l. 29 'competition between sympatric species' is mentioned only once more, on p. 11, l. 40. The paper presents no concrete evidence to support it.

Introduction

p. 3, l.19 What does exceptional mean here? Are all articulated specimens exceptional, in which case the word is redundant?

l. 27 'The diversity of carapace forms among hurdiids has the potential to provide untapped ecological and evolutionary insight into this group. A broader quantitative understanding of carapace shape variation and its potential functional and ecological significance has therefore become necessary.' Unfortunately the paper does not deliver on this ambition (see comment on p. 12, l. 5).

l. 31 'Relationship of carapace shape to aspects of ecology such as feeding' but, unfortunately, you don't use appendage morphology as primary data, but rely on interpretations of function.

p. 5, l. 31 Why only a 'probable' moult assemblage? What is the evidence that it could be a carcass?

l. 36 'Stages' implies a sequence - perhaps 'styles'.

p. 6, l. 3 In what sense are these 'sub'-localities?

l. 4 The Systematic section of the paper, which includes the erection of a new genus and species, provides no statement of the justification for doing this. In what sense does this radiodont differ from *Cambroraster* to a degree sufficient to warrant a new genus? Where are the small specimens of *Megaraster* - among those of *Cambroraster*? The morphometric analysis provides some answers but the comments in the Discussion at the end of the paper are different in their focus.

l. 11 'is the best example for descriptive purposes' (but not the holotype). There is an anomaly here perhaps reflecting the focus on carapace shape. This is a systematic description of a new taxon so why select a specimen for holotype which, by inference, is NOT the best example for descriptive purposes?

The Description includes significant text devoted to comparison with other radiodont carapaces which is not an appropriate part of this section and should be moved to the Discussion.

p. 7, l. 15 The evidence for 'clear signs of deformation' is not evident on the illustration so the reader is left wondering whether the argument relies only on the unusual outline (which would

involve an element of circularity). The is also critical because the H and P elements on the slab do not obviously fit together.

l. 20 'flattened' here appears to refer to the outline in which case it's not the best descriptor due to the potential for confusion with compression artefacts.

l. 29 But the P elements in Hurdia extend well below the margins of the H element as reconstructed by Daley et al. Here (Fig. 5). Here, however, they are enclosed by the H element and it is difficult to interpret their function.

l. 32 'poorly preserved pair of frontal appendages'. Does this affect attempts to interpret the ecology of Megaraster?

p.7, l. 34 'discerned' rather than 'gathered'.

l. 36 They don't look 'needle-like' to me. That would imply a narrower and parallel sided outline.

p. 8, l. 3 The heading and text of Section 3.6 should make it clear that the reinterpretation of Pahvantia concerns the appendage morphology (and function), not the identification.

p. 12, l. 5-22 This paragraph concludes that carapace morphology does not strongly correlate with interpretations of ecology. This inference is likely correct, but the shortcoming of this approach is that the comparison is with designated ecological categories and not morphological characters of the frontal appendage. The consequence is a number of highly qualified statements such as 'together these observations would seem to speak against', 'it is possible that ... but could also be', 'might have played a part'. While such caution is appropriate, this discussion, and the paper as a whole, would be much stronger if the relationship between carapace characters and appendage characters was explored in a PCA or similar, directly. As it stands the paper does not deliver on its goal of providing a 'broader quantitative understanding of carapace shape variation and its potential functional and ecological significance'.

Fig. 5 dashed lines are not obvious (or evident?)

Review form: Reviewer 2

Is the manuscript scientifically sound in its present form?

Yes

Are the interpretations and conclusions justified by the results?

Yes

Is the language acceptable?

Yes

Do you have any ethical concerns with this paper?

No

Have you any concerns about statistical analyses in this paper?

No

Recommendation?

Major revision is needed (please make suggestions in comments)

Comments to the Author(s)

This manuscript by Caron and Moysiuk presents a new radiodont taxon from the Burgess Shale. This is an important fossil locality preserving exceptionally preserved fossils from the Cambrian Explosion, which have proved invaluable for understanding the early evolution of animals. The fossils in this manuscript are beautifully photographed and well described, followed by geometric morphometric analyses and phylogenetic analyses. In general, this is an excellent manuscript that is full of new fossil data and rigorous analyses, and the conclusions of the paper for the most part are well supported by the data. All data and analyses have been provided in the Supplementary Material, and seems to be done to a high standard. I have two major points related to taxonomy and systematics that must be considered before the paper can be published, followed by a series of suggestions for minor to moderate changes. I believe that this paper will be a valuable and interesting contribution to the field once these points have been addressed.

Major comments: Two considerations of Systematics and Taxonomy.

(1) There is no provided justification for *Megaraster* being a new genus. The authors could seriously consider making this taxon a new species of *Cambroraster* rather than a new genus. Although the H-elements are different in size and details of the anatomy, they have an overall broad similarity to the outline. The frontal appendage morphology seems to match that of *Cambroraster* (except for a possible difference in length of distal endites), and the P-elements are not that different either – in any case both these features are not extremely well known in the new taxon. *Hurdia victoria* and *Hurdia triangulata* have rather different H-elements but were left in the same genus because of the similarity of their other body parts. It is unclear why the authors here consider this taxon to be a new genus, and not just a new species of *Cambroraster*. This must be justified much more clearly and convincingly in the text. Currently the *Megaraster* diagnosis only includes features of the H-element, and to me this is not enough to justify a new genus. The phylogeny argument (that they are not sister taxa) is weak in this case because the phylogeny is heavily influenced by how the authors code the taxa, and also because there is no resolution between *Megaraster* and *Cordaticaris*, and *Zhenghecaris* is poorly known. Ultimately the support values for the nodes in this part of the tree are very weak. I think it remains a very real possibility that the new taxon and *Cambroraster* are one genus and two species. This needs to be addressed in the revision.

(2) The redescription of *Pahvantia* provided by the authors is brilliant, and seems very solid. The information from the counterpart convincingly demonstrates that *Pahvantia* seems to have a very similar frontal appendage to *Hurdia*, as the authors show very clearly in figure 6a-d. I would strongly urge the authors to go even further with this redescription, and consider synonymising *Pahvantia* with *Hurdia*. Given the similarity in setal blades, frontal appendage, and P-element, *Pahvantia hastata* may be best considered as a species of *Hurdia* (ie. *Hurdia hastata*). The difference between *Hurdia victoria* and *Hurdia triangulata* is essentially the different morphology of the H-element, which is also now the case with “*Pahvantia*”. Broadly speaking, the H-elements aren’t that different between “*Pahvantia*” and *Hurdia*. The text needs to include an assessment of what the new interpretation means for the systematics of this taxon (even if you don’t agree with my suggestion to synonymise with *Hurdia*).

Minor to moderate comments:

Abstract, page 1 line 18: when describing the “variety of ecological niches” only “benthic foragers to agile nektonic apex predators”. Suspension feeding is not listed and could be added.

Abstract, page 1 lines 26-17: The paper seems to conclude that the variation in shape of the carapace is not really related to feeding ecology, but the abstract here says that the “carapace

shape is prone to homoplasy, likely due to a complex and varied relationship with ecology.” This doesn’t seem to reflect correctly the conclusions made in the discussion of the paper.

Introduction, page 2, line 2: Not all radiodonts have an oral cone strictly speaking, because of taxa such as *Amplectobelua* and *Ramskoeldia* and their GLSs (although these do have toothed plates, just not arranged into an oral cone).

Introduction, page 2, line 18: Add reference 1 to the end of the second paragraph, as Daley et al. 2009 were the first to recognise *Hurdia* as a radiodont and describe it as having a large carapace complex.

Introduction, page 2, lines 19-20: Hurdiids were not known to be radiodonts before 2009. So here when the text says “many described over the past decade” it doesn’t quite correctly describe the history. Nearly all of them were described over the past decade, with the exception of *Peytoia*, and the concept of hurdiids as a family was only developed within the last decade. Again, reference 1 should be cited here.

Materials and methods, page 3, line 16: correct to “In some case, they appear yellowish...” (“they” was missing)

Materials and methods, page 3 line 24 to page 4 line 2: The description of the geometric morphometrics methods is a bit unclear and too brief here. The landmark analysis methods don’t seem to be complete – there is no statement about what the PCA analysis is based on (presumably the Procrustes coordinates?). On line 30, the Elliptical Fourier Analysis was presumably conducted on outlines, rather than landmarks, but this isn’t actually stated. This text here also gets a bit confusing, because it seems at first reading that the PCA was only conducted on the EFA. It’s only after going through the Supplementary Information (which is also unclear and lacking in places) that I eventually realised that PCA was done for both the landmark and outline datasets. In general, this section of the methods could be expanded in the main text to make a clearer and more detailed description of the morphometrics. The ontogenetic allometry test text doesn’t say which of the datasets (landmark or outline) was used for this part of the methods, and again is very brief, with no real explanation in the supplementary material.

Materials and methods, page 4, lines 3 to 15: The description of the phylogenetic methods are also too brief and quite unclear. The text here introduces two approaches that produce the “D tree” and the “L tree” and the “N tree”, and says that the details are in the supplementary material, however there is no mention of these three trees using the same terminology. This is quite confusing. It is also not adequately explained if the D tree method (five discrete characters) and the L tree method (incorporating species mean shape data) were both done together, or as two separately analyses. If they were done together, it seems like that would be redundant. There are also questions that remain about the details of the methods used, even when reading what was supplied in the supplementary materials. The main text for the D tree method should make it clear that characters from previous analyses were removed, and replaced with new characters in the matrix. The supplementary materials description of the L tree method is unclear. What does it mean to “calculate the mean shape for each species”? What about the variation in shape seen in each species? Likewise the methods for the phylomorphospace are very brief, and should be expanded for clarity.

Materials and methods, page 4, lines 16-18: The boxplots were constructed with length measurements only, although elsewhere in the text is mentions that the width of largest *Cambroraster* specimen exceeds the width of *Megaraster* (page 10 line 25). The data for width has not been provided. Since this is one of the arguments for these two taxa not being the same

species, those width data should be provided and I would suggest include a boxplot for width alongside the boxplot for length in figure 9.

Fossil description, page 4, line 19: It would be clearer if this Fossil Description section had two main sections: 3.1 Systematic Paleontology (for *Megaraster*) and 3.2 Reinterpretation of *Pahvantia hastata*. At the moment it almost reads like *Pahvantia hastata* is included as part of *Megaraster* because the Superphylum, Order, Family headings apply to the entire Fossil Description section including part 3.6.

Fossil description, page 4, line 25-26: The name *Megaraster* sounds like it is referring to the large size of the rake-like appendage (rather than the large size of the central carapace element). Considering that the appendage is relatively poorly known in *Megaraster* and it is not included in the diagnosis, maybe a different name could be considered that refers to the large size of the carapace (if the authors decide to keep it as a separate genus, but see my major comment about including it as a new species of *Cambroraster*).

Fossil description, pages 5-6: The anatomy of the new taxon morphology is in some places hard to link to the figures provided. When describing anatomical features in the text, refer to the associated acronym labelling the feature in the figure. So for example, when describing ocular notches, refer to "On in figure 1a" instead of just "figure 1a". Do this throughout the description in order to make it much easier for the reader to link the written description of the specimens.

Fossil description, page 5, lines 22 to 29: This insightful text identifying the anterolateral processes in *Hurdia* seems like it would better fit in the discussion than here.

Fossil description, page 6, lines 32 to 42: Description of the parts of the body other than the carapaces is somewhat lacking in details. The length, shape and paired (or unpaired) nature of the secondary spines of the frontal appendage endites could be described in more detail than just "needle-like". The reference to "enditic spines" is completely unclear – are these different endites to those already described, are they spines on the endites or different endites themselves... etc. How are they different from the other endites, and are they longer or shorter than the other endites, how many are there, etc. These "enditic spines" looks like they have roughly the same relative length compared to the podomeres as in *Cambroraster*, contrary to what is written in the text.

Fossil description, page 7, line 39: The authors here interpret the peduncle of *Pahvantia* as just the semi-circular structure, whereas in Lerosey-Aubril & Pates they included the elongated narrow region that extends dorsally upwards from what these authors consider as the base of the peduncle. Some comment as to why the authors do not consider this material to be part of the appendage, contrary to the original description, and what they think this structure is, should be included.

Results Phylogeny, page 9, line 34: The placement of *Schinderhannes* as the most basal hurdiid seems strange given it is so much younger than all the other radiodonts. But I expect that the basal part of the hurdiids clade would be very much changed if the taxon *Ursulinacaris* had been included in the analysis, because of the presence in that hurdiid of paired endites (Pates et al. 2019 *Zoological Letters*).

Results Phylogeny, page 9, lines 24-27: I think this text here somewhat overstates the helpfulness of carapace characters to the phylogeny. Yes, some of the hurdiids are slightly better resolved, but this comes at the great expense of further destroying resolution further down the tree. Everything other than a hurdiid is just a rake and totally unresolved. This is not ideal. In general, the

importance of the phylogenetic results are quite overstated and overanalysed in the discussion (see my further comments below).

Results Phylogeny, page 10, line 6: Refer more clearly to the “phyломorphospace” here rather than the “phylogeny”. There is not overall description of the phylormorphospace results provided here in the results text. Some of the results are presented later in the manuscript, in the discussion (page 11, lines 5-22). A description of the results from both phylomorphospace analyses should be added to the results.

Discussion, page 10, line 13: Here the authors claim that the carapace complex of the new taxon is “significantly” larger than the largest specimens from other Cambrian hurdiids, but no statistical significance text was reported in the results. Also, it was stated elsewhere in the text that the width of some Cambroraster specimens is larger than the width of the next taxon.

Discussion, page 10, line 20: It is unclear why reference 22 is listed here as giving a size for Anomalocaris, when that is not found in the publication. Reference 12 is better for giving the body size of this taxon.

Discussion, page 10, lines 22-29: I agree with this text that the new taxon and Cambroraster are not the same species, however no justification has been provided as to why it needs to be a new genus and not a species of Cambroraster. The phylogeny argument (that they are not sister taxa) is weak in this case because the phylogeny is heavily influenced by how the authors code the taxa, and also because there is no resolution between Megaraster and Cordaticaris, and Zhenghecaris is poorly known. Ultimately the support values for the nodes in this part of the tree are very weak. I think it remains a very real possibility that the new taxon and Cambroraster are one genus and two species. This needs to be addressed in the revision.

Discussion, page 10 lines 30 to page 11 line 43: The entire discussion about ecology and phylogeny is confusing, circular and difficult to follow. All of this text needs revision. Ultimately, the results show that carapace shape doesn't really have much of an effect on the phylogenetic relationships found in the analysis. This is shown by the phylomorphospace where the things that are closely related on the phylogeny are not plotting near each other in the morphospace. In contrast, feeding ecology (inferred from appendages and nothing to do with carapaces) is quite nicely resolved in the phylogeny. In the discussion text of this paper, the authors seem to confuse and mix up specific feeding ecology derived from interpretation of appendages, with general ecology such as life habit, swimming ability, etc. It goes from saying that there is no “relationship between carapace shape with prey size niche or associated habitat” (page 11 lines 19-20) to saying the opposite that “enlargement of carapaces represent a consequence of adaptation to an increasingly specialized, typically benthic niche linked directly to particular feeding ecology but could also be the side effect of other factors such as the necessity of evolving larger carapaces for defense” (page 11 lines 33-36) and then goes back to saying there is a “lack of evidence for a general correlation between carapace shape and feeding ecology” (page 11 lines 37-38) – this just seems to flip flop back and forth about the link or not between carapace morphology and ecology. This part of the text can probably be made much more concise.

Discussion, page 11, line 23: It is unclear exactly what taxa are meant with “aforementioned hurdiid taxa”.

Discussion, page 11, line 26: For the taxa listed as having “smaller carapace complexes” it should be considered that the carapace complex of Stanleycaris is not known so we can't say if it is small or large. Peytoia has evidence for a carapace that covered all around the front of the head and back as far as the eyes at least. While this doesn't seem to extend forward of the body, it certainly isn't small either. The cephalic carapace situation in Schinderhannes is also very poorly known.

Discussion, page 11, line 27: Is the “more elongate habitus” referring to the appendage or the podomeres or what specifically?

Discussion, page 11, line 29: The reference here to ref. 2 is not correct, as that paper did not describe anything to do with the three taxa being considered here. Please reference the papers that actually discuss and describe the morphology of these appendages, such as Kuhl et al. 2009; Daley et al. 2009, 2013; Caron et al. 2010, etc.

Discussion, page 11, lines 23 to 32: When discussing the evolutionary trends within hurdiids, the taxon *Ursulinacaris* is quite relevant and should be included in the discussion (Pates et al. 2019 *Zoological Letters*).

Figure 1: The white space between the figure panels could be made a bit thicker to make the distinction between the different photos clearer.

Figure 2: “Fa” is missing from caption.

Figure 3: The label “Sp” is not described in any figure caption. Add a figure panel letter to the drawing and include in the text caption as a separate part of the figure. Move the labels from the photograph of the appendages and oral cone to the drawing of those features. Label the boxes in (a) with the letter label of the closeup in the figure.

Figure 5: It seems unnecessary to have four different angles provided for the reconstruction, when the body is not known and the arrangement of the body parts is derived from other radiodonts. Maybe just a single reconstruction that focuses on the head region would be enough. The rest is just clearly copied from *Cambroraster* and adds nothing of relevance to the paper.

Figure 6: There is also a feature labelled as “Sp” in figure 6c. Is this the same feature as the unexplained “Sp” that is found in Figure 3? If not, give them different acronyms.

Figure 7: Add (a) and (b) for the left and right parts of this figure. In the right figure, label the key taxa *Zhenghecaris* and *Aegirocassis*. The purple colour of *Aegirocassis* is extremely difficult to distinguish from the black dots.

Figure 8: In (a), remove the “Hurdiinae” line, as this is never discussed or mentioned in the manuscript text.

Decision letter (RSOS-202138.R0)

Dear Dr Caron

The Editors assigned to your paper RSOS-202138 "A giant nektobenthic radiodont from the Burgess Shale and the significance of hurdiid carapace diversity" have made a decision based on their reading of the paper and any comments received from reviewers.

Regrettably, in view of the reports received, the manuscript has been rejected in its current form. However, a new manuscript may be submitted which takes into consideration these comments.

I note that both reviewers and the Associate Editor see very positive aspects of this manuscript. However the reviewers between raise a large set of significant reservations about the work, all of which seem potentially valid and require your attention. To bring the manuscript to a form where the reviewers' concerns are resolved will take significant work and time -- substantially more than is typically required for a 'major revision'. The decision of rejection with resubmission allowed has been taken to emphasise this point. I hope that in due course you will be able to make a suitable resubmission.

We invite you to respond to the comments supplied below and prepare a resubmission of your manuscript. Below the referees' and Editors' comments (where applicable) we provide additional requirements. We provide guidance below to help you prepare your revision.

Please note that resubmitting your manuscript does not guarantee eventual acceptance, and we do not generally allow multiple rounds of revision and resubmission, so we urge you to make every effort to fully address all of the comments at this stage. If deemed necessary by the Editors, your manuscript will be sent back to one or more of the original reviewers for assessment. If the original reviewers are not available, we may invite new reviewers.

Please resubmit your revised manuscript and required files (see below) no later than 10-Aug-2021. Note: the ScholarOne system will 'lock' if resubmission is attempted on or after this deadline. If you do not think you will be able to meet this deadline, please contact the editorial office immediately.

Please note article processing charges apply to papers accepted for publication in Royal Society Open Science (<https://royalsocietypublishing.org/rsos/charges>). Charges will also apply to papers transferred to the journal from other Royal Society Publishing journals, as well as papers submitted as part of our collaboration with the Royal Society of Chemistry (<https://royalsocietypublishing.org/rsos/chemistry>). Fee waivers are available but must be requested when you submit your manuscript (<https://royalsocietypublishing.org/rsos/waivers>).

Thank you for submitting your manuscript to Royal Society Open Science and we look forward to receiving your resubmission. If you have any questions at all, please do not hesitate to get in touch.

on behalf of Professor Elizabeth Harper (Associate Editor) and Peter Haynes (Subject Editor)
openscience@royalsociety.org

Associate Editor Comments to Author (Professor Elizabeth Harper):

Associate Editor: 1

Comments to the Author:

This could be an extremely interesting contribution and it is obvious that the material has significance.

However, the two reviewers point out between them very real problems with the ms (lack of clear of reason to erect a new genus, poorly described phylogenetic analysis, poor (and

sometimes conflicting) links made between morphology and ecology. I would encourage you to address these issues before resubmission. The reviewers make numerous helpful remarks - these should all be addressed point by point.

Reviewer comments to Author:

Reviewer: 1

Comments to the Author(s)

p. 2, l.21 The Abstract refers to 'sediment-sifting' yet the first paragraph describes hurdiids as 'specialized for sweep feeding ... in the water column'?

l.22 'zeppelin-shaped' is used in the Abstract and nowhere else (i.e. not in the Description). The outline of the carapace does not resemble a zeppelin.

l. 26 'a complex and varied relationship with ecology' is no doubt true, but it's also a euphemism for a lack of results.

l. 29 'competition between sympatric species' is mentioned only once more, on p. 11, l. 40. The paper presents no concrete evidence to support it.

Introduction

p. 3, l.19 What does exceptional mean here? Are all articulated specimens exceptional, in which case the word is redundant?

l. 27 'The diversity of carapace forms among hurdiids has the potential to provide untapped ecological and evolutionary insight into this group. A broader quantitative understanding of carapace shape variation and its potential functional and ecological significance has therefore become necessary.' Unfortunately the paper does not deliver on this ambition (see comment on p. 12, l. 5).

l. 31 'Relationship of carapace shape to aspects of ecology such as feeding' but, unfortunately, you don't use appendage morphology as primary data, but rely on interpretations of function.

p. 5, l. 31 Why only a 'probable' moult assemblage? What is the evidence that it could be a carcass?

l. 36 'Stages' implies a sequence - perhaps 'styles'.

p. 6, l. 3 In what sense are these 'sub'-localities?

l. 4 The Systematic section of the paper, which includes the erection of a new genus and species, provides no statement of the justification for doing this. In what sense does this radiodont differ from *Cambroraster* to a degree sufficient to warrant a new genus? Where are the small specimens of *Megaraster* - among those of *Cambroraster*? The morphometric analysis provides some answers but the comments in the Discussion at the end of the paper are different in their focus.

l. 11 'is the best example for descriptive purposes' (but not the holotype). There is an anomaly here perhaps reflecting the focus on carapace shape. This is a systematic description of a new taxon so why select a specimen for holotype which, by inference, is NOT the best example for descriptive purposes?

The Description includes significant text devoted to comparison with other radiodont carapaces which is not an appropriate part of this section and should be moved to the Discussion.

p. 7, l. 15 The evidence for 'clear signs of deformation' is not evident on the illustration so the reader is left wondering whether the argument relies only on the unusual outline (which would involve an element of circularity). The is also critical because the H and P elements on the slab do not obviously fit together.

l. 20 'flattened' here appears to refer to the outline in which case it's not the best descriptor due to the potential for confusion with compression artefacts.

l. 29 But the P elements in *Hurdia* extend well below the margins of the H element as reconstructed by Daley et al. Here (Fig. 5). Here, however, they are enclosed by the H element and it is difficult to interpret their function.

l. 32 'poorly preserved pair of frontal appendages'. Does this affect attempts to interpret the ecology of *Megaraster*?

p.7, l. 34 'discerned' rather than 'gathered'.

l. 36 They don't look 'needle-like' to me. That would imply a narrower and parallel sided outline.

p. 8, l. 3 The heading and text of Section 3.6 should make it clear that the reinterpretation of *Pahvantia* concerns the appendage morphology (and function), not the identification.

p. 12, l. 5-22 This paragraph concludes that carapace morphology does not strongly correlate with interpretations of ecology. This inference is likely correct, but the shortcoming of this approach is that the comparison is with designated ecological categories and not morphological characters of the frontal appendage. The consequence is a number of highly qualified statements such as 'together these observations would seem to speak against', 'it is possible that ... but could also be', 'might have played a part'. While such caution is appropriate, this discussion, and the paper as a whole, would be much stronger if the relationship between carapace characters and appendage characters was explored in a PCA or similar, directly. As it stands the paper does not deliver on its goal of providing a 'broader quantitative understanding of carapace shape variation and its potential functional and ecological significance'.

Fig. 5 dashed lines are not obvious (or evident?)

Reviewer: 2

Comments to the Author(s)

This manuscript by Caron and Moysiuk presents a new radiodont taxon from the Burgess Shale. This is an important fossil locality preserving exceptionally preserved fossils from the Cambrian Explosion, which have proved invaluable for understanding the early evolution of animals. The fossils in this manuscript are beautifully photographed and well described, followed by geometric morphometric analyses and phylogenetic analyses. In general, this is an excellent manuscript that is full of new fossil data and rigorous analyses, and the conclusions of the paper for the most part are well supported by the data. All data and analyses have been provided in the Supplementary Material, and seems to be done to a high standard. I have two major points related to taxonomy and systematics that must be considered before the paper can be published, followed by a series of suggestions for minor to moderate changes. I believe that this paper will be a valuable and interesting contribution to the field once these points have been addressed.

Major comments: Two considerations of Systematics and Taxonomy.

(1) There is no provided justification for *Megaraster* being a new genus. The authors could seriously consider making this taxon a new species of *Cambroraster* rather than a new genus. Although the H-elements are different in size and details of the anatomy, they have an overall broad similarity to the outline. The frontal appendage morphology seems to match that of *Cambroraster* (except for a possible difference in length of distal endites), and the P-elements are not that different either – in any case both these features are not extremely well known in the new taxon. *Hurdia victoria* and *Hurdia triangulata* have rather different H-elements but were left in the same genus because of the similarity of their other body parts. It is unclear why the authors here consider this taxon to be a new genus, and not just a new species of *Cambroraster*. This must be justified much more clearly and convincingly in the text. Currently the *Megaraster* diagnosis only includes features of the H-element, and to me this is not enough to justify a new genus. The phylogeny argument (that they are not sister taxa) is weak in this case because the phylogeny is heavily influenced by how the authors code the taxa, and also because there is no resolution between *Megaraster* and *Cordaticaris*, and *Zhenghecaris* is poorly known. Ultimately the support values for the nodes in this part of the tree are very weak. I think it remains a very real possibility that the new taxon and *Cambroraster* are one genus and two species. This needs to be addressed in the revision.

(2) The redescription of *Pahvantia* provided by the authors is brilliant, and seems very solid. The information from the counterpart convincingly demonstrates that *Pahvantia* seems to have a very similar frontal appendage to *Hurdia*, as the authors show very clearly in figure 6a-d. I would strongly urge the authors to go even further with this redescription, and consider synonymising *Pahvantia* with *Hurdia*. Given the similarity in setal blades, frontal appendage, and P-element, *Pahvantia hastata* may be best considered as a species of *Hurdia* (ie. *Hurdia hastata*). The difference between *Hurdia victoria* and *Hurdia triangulata* is essentially the different morphology of the H-element, which is also now the case with “*Pahvantia*”. Broadly speaking, the H-elements aren’t that different between “*Pahvantia*” and *Hurdia*. The text needs to include an assessment of what the new interpretation means for the systematics of this taxon (even if you don’t agree with my suggestion to synonymise with *Hurdia*).

Minor to moderate comments:

Abstract, page 1 line 18: when describing the “variety of ecological niches” only “benthic foragers to agile nektonic apex predators”. Suspension feeding is not listed and could be added.

Abstract, page 1 lines 26-17: The paper seems to conclude that the variation in shape of the carapace is not really related to feeding ecology, but the abstract here says that the “carapace shape is prone to homoplasy, likely due to a complex and varied relationship with ecology.” This doesn’t seem to reflect correctly the conclusions made in the discussion of the paper.

Introduction, page 2, line 2: Not all radiodonts have an oral cone strictly speaking, because of taxa such as *Amplectobelua* and *Ramskoeldia* and their GLSs (although these do have toothed plates, just not arranged into an oral cone).

Introduction, page 2, line 18: Add reference 1 to the end of the second paragraph, as Daley et al. 2009 were the first to recognise *Hurdia* as a radiodont and describe it as having a large carapace complex.

Introduction, page 2, lines 19-20: Hurdiids were not known to be radiodonts before 2009. So here when the text says “many described over the past decade” it doesn’t quite correctly describe the history. Nearly all of them were described over the past decade, with the exception of *Peytoia*, and the concept of hurdiids as a family was only developed within the last decade. Again, reference 1 should be cited here.

Materials and methods, page 3, line 16: correct to “In some case, they appear yellowish...” (“they” was missing)

Materials and methods, page 3 line 24 to page 4 line 2: The description of the geometric morphometrics methods is a bit unclear and too brief here. The landmark analysis methods don't seem to be complete – there is no statement about what the PCA analysis is based on (presumably the Procrustes coordinates?). On line 30, the Elliptical Fourier Analysis was presumably conducted on outlines, rather than landmarks, but this isn't actually stated. This text here also gets a bit confusing, because it seems at first reading that the PCA was only conducted on the EFA. It's only after going through the Supplementary Information (which is also unclear and lacking in places) that I eventually realised that PCA was done for both the landmark and outline datasets. In general, this section of the methods could be expanded in the main text to make a clearer and more detailed description of the morphometrics. The ontogenetic allometry test text doesn't say which of the datasets (landmark or outline) was used for this part of the methods, and again is very brief, with no real explanation in the supplementary material.

Materials and methods, page 4, lines 3 to 15: The description of the phylogenetic methods are also too brief and quite unclear. The text here introduces two approaches that produce the “D tree” and the “L tree” and the “N tree”, and says that the details are in the supplementary material, however there is no mention of these three trees using the same terminology. This is quite confusing. It is also not adequately explained if the D tree method (five discrete characters) and the L tree method (incorporating species mean shape data) were both done together, or as two separately analyses. If they were done together, it seems like that would be redundant. There are also questions that remain about the details of the methods used, even when reading what was supplied in the supplementary materials. The main text for the D tree method should make it clear that characters from previous analyses were removed, and replaced with new characters in the matrix. The supplementary materials description of the L tree method is unclear. What does it mean to “calculate the mean shape for each species”? What about the variation in shape seen in each species? Likewise the methods for the phylomorphospace are very brief, and should be expanded for clarity.

Materials and methods, page 4, lines 16-18: The boxplots were constructed with length measurements only, although elsewhere in the text is mentions that the width of largest Cambroraster specimen exceeds the width of Megaraster (page 10 line 25). The data for width has not been provided. Since this is one of the arguments for these two taxa not being the same species, those width data should be provided and I would suggest include a boxplot for width alongside the boxplot for length in figure 9.

Fossil description, page 4, line 19: It would be clearer if this Fossil Description section had two main sections: 3.1 Systematic Paleontology (for Megaraster) and 3.2 Reinterpretation of Pahvantia hastata. At the moment it almost reads like Pahvantia hastata is included as part of Megaster because the Superphylum, Order, Family headings apply to the entire Fossil Description section including part 3.6.

Fossil description, page 4, line 25-26: The name Megaraster sounds like it is referring to the large size of the rake-like appendage (rather than the large size of the central carapace element). Considering that the appendage is relatively poorly known in Megaraster and it is not included in the diagnosis, maybe a different name could be considered that refers to the large size of the carapace (if the authors decide to keep it as a separate genus, but see my major comment about including it as a new species of Cambroraster).

Fossil description, pages 5-6: The anatomy of the new taxon morphology is in some places hard to link to the figures provided. When describing anatomical features in the text, refer to the associated acronym labelling the feature in the figure. So for example, when describing ocular notches, refer to “On in figure 1a” instead of just “figure 1a”. Do this throughout the description in order to make it much easier for the reader to link the written description of the specimens.

Fossil description, page 5, lines 22 to 29: This insightful text identifying the anterolateral processes in *Hurdia* seems like it would better fit in the discussion than here.

Fossil description, page 6, lines 32 to 42: Description of the parts of the body other than the carapaces is somewhat lacking in details. The length, shape and paired (or unpaired) nature of the secondary spines of the frontal appendage endites could be described in more detail than just “needle-like”. The reference to “enditic spines” is completely unclear – are these different endites to those already described, are they spines on the endites or different endites themselves... etc. How are they different from the other endites, and are they longer or shorter than the other endites, how many are there, etc. These “enditic spines” looks like they have roughly the same relative length compared to the podomeres as in *Cambroaster*, contrary to what is written in the text.

Fossil description, page 7, line 39: The authors here interpret the peduncle of *Pahvantia* as just the semi-circular structure, whereas in *Lerosey-Aubril & Pates* they included the elongated narrow region that extends dorsally upwards from what these authors consider as the base of the peduncle. Some comment as to why the authors do not consider this material to be part of the appendage, contrary to the original description, and what they think this structure is, should be included.

Results Phylogeny, page 9, line 34: The placement of *Schinderhannes* as the most basal hurdiid seems strange given it is so much younger than all the other radiodonts. But I expect that the basal part of the hurdiids clade would be very much changed if the taxon *Ursulinacaris* had been included in the analysis, because of the presence in that hurdiid of paired endites (*Pates et al. 2019 Zoological Letters*).

Results Phylogeny, page 9, lines 24-27: I think this text here somewhat overstates the helpfulness of carapace characters to the phylogeny. Yes, some of the hurdiids are slightly better resolved, but this comes at the great expense of further destroying resolution further down the tree. Everything other than a hurdiid is just a rake and totally unresolved. This is not ideal. In general, the importance of the phylogenetic results are quite overstated and overanalysed in the discussion (see my further comments below).

Results Phylogeny, page 10, line 6: Refer more clearly to the “phylogenetic space” here rather than the “phylogeny”. There is not overall description of the phylogenetic space results provided here in the results text. Some of the results are presented later in the manuscript, in the discussion (page 11, lines 5-22). A description of the results from both phylogenetic space analyses should be added to the results.

Discussion, page 10, line 13: Here the authors claim that the carapace complex of the new taxon is “significantly” larger than the largest specimens from other Cambrian hurdiids, but no statistical significance text was reported in the results. Also, it was stated elsewhere in the text that the width of some *Cambroaster* specimens is larger than the width of the next taxon.

Discussion, page 10, line 20: It is unclear why reference 22 is listed here as giving a size for *Anomalocaris*, when that is not found in the publication. Reference 12 is better for giving the body size of this taxon.

Discussion, page 10, lines 22-29: I agree with this text that the new taxon and *Cambroraster* are not the same species, however no justification has been provided as to why it needs to be a new genus and not a species of *Cambroraster*. The phylogeny argument (that they are not sister taxa) is weak in this case because the phylogeny is heavily influenced by how the authors code the taxa, and also because there is no resolution between *Megaraster* and *Cordaticaris*, and *Zhenghecaris* is poorly known. Ultimately the support values for the nodes in this part of the tree are very weak. I think it remains a very real possibility that the new taxon and *Cambroraster* are one genus and two species. This needs to be addressed in the revision.

Discussion, page 10 lines 30 to page 11 line 43: The entire discussion about ecology and phylogeny is confusing, circular and difficult to follow. All of this text needs revision. Ultimately, the results show that carapace shape doesn't really have much of an effect on the phylogenetic relationships found in the analysis. This is shown by the phylomorphospace where the things that are closely related on the phylogeny are not plotting near each other in the morphospace. In contrast, feeding ecology (inferred from appendages and nothing to do with carapaces) is quite nicely resolved in the phylogeny. In the discussion text of this paper, the authors seem to confuse and mix up specific feeding ecology derived from interpretation of appendages, with general ecology such as life habit, swimming ability, etc. It goes from saying that there is no "relationship between carapace shape with prey size niche or associated habitat" (page 11 lines 19-20) to saying the opposite that "enlargement of carapaces represent a consequence of adaptation to an increasingly specialized, typically benthic niche linked directly to particular feeding ecology but could also be the side effect of other factors such as the necessity of evolving larger carapaces for defense" (page 11 lines 33-36) and then goes back to saying there is a "lack of evidence for a general correlation between carapace shape and feeding ecology" (page 11 lines 37-38) – this just seems to flip flop back and forth about the link or not between carapace morphology and ecology. This part of the text can probably be made much more concise.

Discussion, page 11, line 23: It is unclear exactly what taxa are meant with "aforementioned hurdiid taxa".

Discussion, page 11, line 26: For the taxa listed as having "smaller carapace complexes" it should be considered that the carapace complex of *Stanleycaris* is not known so we can't say if it is small or large. *Peytoia* has evidence for a carapace that covered all around the front of the head and back as far as the eyes at least. While this doesn't seem to extend forward of the body, it certainly isn't small either. The cephalic carapace situation in *Schinderhannes* is also very poorly known.

Discussion, page 11, line 27: Is the "more elongate habitus" referring to the appendage or the podomeres or what specifically?

Discussion, page 11, line 29: The reference here to ref. 2 is not correct, as that paper did not describe anything to do with the three taxa being considered here. Please reference the papers that actually discuss and describe the morphology of these appendages, such as Kuhl et al. 2009; Daley et al. 2009, 2013; Caron et al. 2010, etc.

Discussion, page 11, lines 23 to 32: When discussing the evolutionary trends within hurdiids, the taxon *Ursulinacaris* is quite relevant and should be included in the discussion (Pates et al. 2019 *Zoological Letters*).

Figure 1: The white space between the figure panels could be made a bit thicker to make the distinction between the different photos clearer.

Figure 2: "Fa" is missing from caption.

Figure 3: The label “Sp” is not described in any figure caption. Add a figure panel letter to the drawing and include in the text caption as a separate part of the figure. Move the labels from the photograph of the appendages and oral cone to the drawing of those features. Label the boxes in (a) with the letter label of the closeup in the figure.

Figure 5: It seems unnecessary to have four different angles provided for the reconstruction, when the body is not known and the arrangement of the body parts is derived from other radiodonts. Maybe just a single reconstruction that focuses on the head region would be enough. The rest is just clearly copied from *Cambroraster* and adds nothing of relevance to the paper.

Figure 6: There is also a feature labelled as “Sp” in figure 6c. Is this the same feature as the unexplained “Sp” that is found in Figure 3? If not, give them different acronyms.

Figure 7: Add (a) and (b) for the left and right parts of this figure. In the right figure, label the key taxa *Zhenghecaris* and *Aegirocassis*. The purple colour of *Aegirocassis* is extremely difficult to distinguish from the black dots.

Figure 8: In (a), remove the “*Hurdiinae*” line, as this is never discussed or mentioned in the manuscript text.

===PREPARING YOUR MANUSCRIPT===

===PREPARING YOUR REVISION IN SCHOLARONE===

Author's Response to Decision Letter for (RSOS-202138.R0)

See Appendix A.

RSOS-210664.R0

Review form: Reviewer 1

Is the manuscript scientifically sound in its present form?

Yes

Are the interpretations and conclusions justified by the results?

Yes

Is the language acceptable?

Yes

Do you have any ethical concerns with this paper?

No

Have you any concerns about statistical analyses in this paper?

No

Recommendation?

Accept with minor revision (please list in comments)

Comments to the Author(s)

The authors have given the comments of both reviewers careful consideration. Where they do not agree with reviewers' suggested changes or incorporate them fully, the differences are issues of interpretation or emphasis. The authors are entitled to their views, which will be available for testing by future researchers.

The Abstract refers to 'sediment-sifting' and the first paragraph still describes hurdiids as 'specialized for sweep feeding ... in the water column'. In their response the authors explain that the two descriptions are not mutually exclusive but the reference to 'water column' is potentially confusing because it can be construed as eliminating feeding on sediment. This can be easily remedied.

Ms p. 2, l. 9. The suggestion that 'these animals occupied different habitats within the water column' appears to conflict with the possibility of competition.

Ms p. 2, l. 27. 'together' is redundant.

Ms p. 3, l. 11. 'preserved as aluminosilicate and carbon films' requires a citation.

Ms p. 5, l. 30. Explain what 'buried flat' means?

Ms p. 6, l. 22. 'specimen' is redundant here and elsewhere. In this case 'holotype' is adequate.

Ms p. 8, l. 17. I suggest 'compatible' rather than 'in tune'.

Ms p. 11, l. 18. reminiscent 'of' not 'to'

Ms p. 12, l. 29. elevates 'to' not 'as'

Review form: Reviewer 2

Is the manuscript scientifically sound in its present form?

Yes

Are the interpretations and conclusions justified by the results?

Yes

Is the language acceptable?

Yes

Do you have any ethical concerns with this paper?

No

Have you any concerns about statistical analyses in this paper?

No

Recommendation?

Accept with minor revision (please list in comments)

Comments to the Author(s)

In the revised version of this manuscript, the authors have adequately addressed all the reviewers comments. The authors have added a lot of new images and drawings of the material, which better illustrate their descriptive work and create a clearer and more coherent foundation for their taxonomic decisions. They also clarified areas of the text to do with the methods and results, making the paper much easier to understand. In general, this revised version is very much improved over the previous version, which was already excellent. I have just a few very minor suggestions below, and I recommend accepting this manuscript once these have been addressed.

Minor suggestions:

With all the changes to the figures, it seems that some call-outs to figures in the text don't accurately correspond anymore with the appropriate image. For example, on page 9 lines 39-40, the text about the holotype appendages and oral cones refers to figure 3b, but this is actually a photo of just the carapace - the authors probably mean to call-out to figure 3a or maybe 3g? All call-outs should be double-checked that they refer to the correct figure.

In figure 3, I suggest that the authors place all labels of anatomy in the drawing in figure 3k, and remove the labels from the associated photograph in figure g. This allows the reader to see more clearly the fossil, without labels getting in the way.

The authors have expanded a bit the statement about a deformed carapace, and figured the original and retrodeformed specimen in figures 3b and 3c. The arrows are not mentioned in the caption, but they seem to point in the opposite direction that the shear force and deformation. I

guess maybe the arrows are showing the direction of retrodeformation, rather than the deformation direction itself? It would make more sense for the arrows to be reversed and show the direction of warping, and also then mention in the caption clearly what the arrows represent. Also on this subject, in the text on page 7 line 27, and page 3 line 32, the authors attribute this deformation to oblique burial of the specimen relative to bedding, however it looks much more like a metamorphic deformation. This is because the agnostids found in the same image are also deformed, with those aligned roughly horizontally looking too flat/thin and those aligned roughly vertical looking to wide/fat. This suggests the whole surface underwent some kind of slight metamorphic warping/deformation, which makes sense given this is in a mountainous region.

On page 7 line 45, the use of the words “finely tapering” to replace some previous text is a bit vague. Does “finely” mean that the taper proceeds gradually along the entire length of the spine, or does it mean that it only tapers right at the end? Replace with a clearer terminology.

Decision letter (RSOS-210664.R0)

Dear Dr Caron

On behalf of the Editors, we are pleased to inform you that your Manuscript RSOS-210664 "A giant nektobenthic radiodont from the Burgess Shale and the significance of hurdiid carapace diversity" has been accepted for publication in Royal Society Open Science subject to minor revision in accordance with the referees' reports. Please find the referees' comments along with any feedback from the Editors below my signature.

Please submit your revised manuscript and required files (see below) no later than 7 days from today's (ie 20-Jul-2021) date. Note: the ScholarOne system will 'lock' if submission of the revision is attempted 7 or more days after the deadline. If you do not think you will be able to meet this deadline please contact the editorial office immediately.

Kind regards,
Royal Society Open Science Editorial Office

on behalf of Professor Elizabeth Harper (Associate Editor) and Peter Haynes (Subject Editor)
 openscience@royalsociety.org

Associate Editor Comments to Author (Professor Elizabeth Harper):

Thank you for your constructive response to the reviews.

Both reviewers seemingly very happy with the work which has gone into clarifying and improving this manuscript. They both make very minor, but useful comments, and these should be attended to before final acceptance.

Reviewer comments to Author:

Reviewer: 1

Comments to the Author(s)

The authors have given the comments of both reviewers careful consideration. Where they do not agree with reviewers' suggested changes or incorporate them fully, the differences are issues of interpretation or emphasis. The authors are entitled to their views, which will be available for testing by future researchers.

The Abstract refers to 'sediment-sifting' and the first paragraph still describes hurdiids as 'specialized for sweep feeding ... in the water column'. In their response the authors explain that the two descriptions are not mutually exclusive but the reference to 'water column' is potentially confusing because it can be construed as eliminating feeding on sediment. This can be easily remedied.

Ms p. 2, l. 9. The suggestion that 'these animals occupied different habitats within the water column' appears to conflict with the possibility of competition.

Ms p. 2, l. 27. 'together' is redundant.

Ms p. 3, l. 11. 'preserved as aluminosilicate and carbon films' requires a citation.

Ms p. 5, l. 30. Explain what 'buried flat' means?

Ms p. 6, l. 22. 'specimen' is redundant here and elsewhere. In this case 'holotype' is adequate.

Ms p. 8, l. 17. I suggest 'compatible' rather than 'in tune'.

Ms p. 11, l. 18. reminiscent 'of' not 'to'

Ms p. 12, l. 29. elevates 'to' not 'as'

Reviewer: 2

Comments to the Author(s)

In the revised version of this manuscript, the authors have adequately addressed all the reviewers comments. The authors have added a lot of new images and drawings of the material, which better illustrate their descriptive work and create a clearer and more coherent foundation for their taxonomic decisions. They also clarified areas of the text to do with the methods and results, making the paper much easier to understand. In general, this revised version is very

much improved over the previous version, which was already excellent. I have just a few very minor suggestions below, and I recommend accepting this manuscript once these have been addressed.

Minor suggestions:

With all the changes to the figures, it seems that some call-outs to figures in the text don't accurately correspond anymore with the appropriate image. For example, on page 9 lines 39-40, the text about the holotype appendages and oral cones refers to figure 3b, but this is actually a photo of just the carapace – the authors probably mean to call-out to figure 3a or maybe 3g? All call-outs should be double-checked that they refer to the correct figure.

In figure 3, I suggest that the authors place all labels of anatomy in the drawing in figure 3k, and remove the labels from the associated photograph in figure g. This allows the reader to see more clearly the fossil, without labels getting in the way.

The authors have expanded a bit the statement about a deformed carapace, and figured the original and retrodeformed specimen in figures 3b and 3c. The arrows are not mentioned in the caption, but they seem to point in the opposite direction that the shear force and deformation. I guess maybe the arrows are showing the direction of retrodeformation, rather than the deformation direction itself? It would make more sense for the arrows to be reversed and show the direction of warping, and also then mention in the caption clearly what the arrows represent. Also on this subject, in the text on page 7 line 27, and page 3 line 32, the authors attribute this deformation to oblique burial of the specimen relative to bedding, however it looks much more like a metamorphic deformation. This is because the agnostids found in the same image are also deformed, with those aligned roughly horizontally looking too flat/thin and those aligned roughly vertical looking to wide/fat. This suggests the whole surface underwent some kind of slight metamorphic warping/deformation, which makes sense given this is in a mountainous region.

On page 7 line 45, the use of the words “finely tapering” to replace some previous text is a bit vague. Does “finely” mean that the taper proceeds gradually along the entire length of the spine, or does it mean that is only tapers right at the end? Replace with a clearer terminology.

===PREPARING YOUR MANUSCRIPT===

While not essential, it will speed up the preparation of your manuscript proof if you format your references/bibliography in Vancouver style (please see

<https://royalsociety.org/journals/authors/author-guidelines/#formatting>). You should include DOIs for as many of the references as possible.

===PREPARING YOUR REVISION IN SCHOLARONE===

<https://royalsociety.org/journals/authors/author-guidelines/#data>. You should ensure that you cite the dataset in your reference list. If you have deposited data etc in the Dryad repository,

please only include the 'For publication' link at this stage. You should remove the 'For review' link.

Author's Response to Decision Letter for (RSOS-210664.R0)

See Appendix B.

Decision letter (RSOS-210664.R1)

Dear Dr Caron,

I am pleased to inform you that your manuscript entitled "A giant nektobenthic radiodont from the Burgess Shale and the significance of hurdiid carapace diversity" is now accepted for publication in Royal Society Open Science.

on behalf of Professor Elizabeth Harper (Associate Editor) and Peter Haynes (Subject Editor)
openscience@royalsociety.org

Appendix A

Associate Editor Comments to Author (Professor Elizabeth Harper):

Associate Editor: 1

Comments to the Author:

This could be an extremely interesting contribution and it is obvious that the material has significance. However, the two reviewers point out between them very real problems with the ms (lack of clear of reason to erect a new genus, poorly described phylogenetic analysis, poor (and sometimes conflicting) links made between morphology and ecology. I would encourage you to address these issues before resubmission. The reviewers make numerous helpful remarks - these should all be addressed point by point.

We are thankful for the interest in our work, and for the helpful remarks. The requested changes were mostly minor and have been easily resolved. We think that there is strong justification for erecting a new genus and we have clarified this and other points in our manuscript. We have made efforts to address all comments below.

Reviewer comments to Author:

Reviewer: 1

Comments to the Author(s)

p. 2, l.21 The Abstract refers to 'sediment-sifting' yet the first paragraph describes hurdiids as 'specialized for sweep feeding ... in the water column'?

We refer to a variety of life modes in different parts of the water column, including benthic regions. Sweep feeding (i.e. sweeping the appendages to trap food items) includes sediment sifting as well as suspension feeding.

l.22 'zeppelin-shaped' is used in the Abstract and nowhere else (i.e. not in the Description). The outline of the carapace does not resemble a zeppelin.

We have replaced this wording by "ovoid-shaped"

l. 26 'a complex and varied relationship with ecology' is no doubt true, but it's also a euphemism for a lack of results.

We have revised the wording here to clarify that no consistent relationship can be demonstrated.

l. 29 'competition between sympatric species' is mentioned only once more, on p. 11, l. 40. The paper presents no concrete evidence to support it.

We agree that no direct evidence can be used to support competition unambiguously, however, since *Titanokorys* and *Cambroraster* have similar appendage-types which are not so different in size and that these two species are preserved together along the same bedding planes, we maintain that this is enough to at least suggest the hypothesis that these two species were potentially competing for similar resources, in particular benthic food sources. We would like to point out that although speculative, such conclusions would be in line with previous suggestions that competition must have played an important role in Cambrian communities. The presence of different great appendage arthropods showing a variety of appendage morphology and sizes, preserved in the same deposits, has been interpreted as the result of the exploitation of specific prey items to reduce competition (e.g. Chen et al. 2004). A similar reasoning coupled with size partitioning was used with the description of various radiodont species preserved in one Burgess Shale deposit (e.g. Daley and Budd, 2010).

Introduction

p. 3, l.19 What does exceptional mean here? Are all articulated specimens exceptional, in which case the word is redundant?

We have removed as suggested.

l. 27 'The diversity of carapace forms among hurdiids has the potential to provide untapped ecological and evolutionary insight into this group. A broader quantitative understanding of carapace shape variation and its potential functional and ecological significance has therefore become necessary.' Unfortunately the paper does not deliver on this ambition (see comment on p. 12, l. 5).

We have reduced this section of the manuscript which was meant more as the presentation of a hypothesis (i.e. Can carapace shape provide valuable ecological and evolutionary insights?). We would like to point out that the discovery of no apparent relationship between shape of carapace and ecology (based on appendage morphology), or the fact that the carapace morphology does not seem to help increase phylogenetic resolutions should be taken cautiously but, in any event, represent results on their own based on the data available at this time. Of course, with additional new data, in particular with more information about appendages of species with poorly known appendages, such as *Cordaticaris* and the discovery of new species, these results might be challenged in the future.

l. 31 'Relationship of carapace shape to aspects of ecology such as feeding' but, unfortunately, you don't use appendage morphology as primary data, but rely on interpretations of function.

We comment on this in response to a similar comment below.

p. 5, l. 31 Why only a 'probable' moult assemblage? What is the evidence that it could be a carcass?

It can be difficult to differentiate between moults and carcasses in fossil arthropods. We think that the consistent organization of elements in the *Titanokorys* assemblage is suggestive of the

former, as also argued for *Cambroraster* previously, but in the absence of a definitive open moulting suture we use the word 'probable.'

l. 36 'Stages' implies a sequence – perhaps 'styles'.

Replaced with states.

p. 6, l. 3 In what sense are these 'sub'-localities?

We have rephrased as localities.

l. 4 The Systematic section of the paper, which includes the erection of a new genus and species, provides no statement of the justification for doing this. In what sense does this radiodont differ from *Cambroraster* to a degree sufficient to warrant a new genus? Where are the small specimens of *Titanokorys* – among those of *Cambroraster*? The morphometric analysis provides some answers but the comments in the Discussion at the end of the paper are different in their focus.

See our detailed response to Referee 2 on related comments. We have clarified the justification for erecting a new genus in the manuscript. The shape of the *Titanokorys* H-element does not align with the reconstructed pattern of allometry found in *Cambroraster* and the two form discrete clusters in shape space. The two also overlap in centroid size (though not length, as *Titanokorys* has a considerably greater L:W ratio). This speaks strongly against an ontogenetic relationship between these forms. We have included further comments in response to referee 2.

l. 11 'is the best example for descriptive purposes' (but not the holotype). There is an anomaly here perhaps reflecting the focus on carapace shape. This is a systematic description of a new taxon so why select a specimen for holotype which, by inference, is NOT the best example for descriptive purposes?

We selected the most complete assemblage as the holotype, this specimen includes not only the H-element but also the paired P-elements, paired appendages, mouth and gill blades. We recognize that this specimen is heavily weathered but still shows many important morphological details, perceptible only with certain low angles of light and we revised Figure accordingly. The H-element was buried obliquely and is asymmetrically deformed. As such our description of the H-element focusses on the paratype, which is better preserved in this respect (but missing all the other elements seen on the holotype). We have clarified this.

The Description includes significant text devoted to comparison with other radiodont carapaces which is not an appropriate part of this section and should be moved to the Discussion.

As suggested, we have moved this into a new comparative section.

p. 7, l. 15 The evidence for 'clear signs of deformation' is not evident on the illustration so the reader is left wondering whether the argument relies only on the unusual outline (which would involve an element of circularity). This is also critical because the H and P elements on the slab do not obviously fit together.

The asymmetric outline and linear compression artefacts are typical of specimens which have been buried obliquely in other Burgess Shale taxa, for example *Hurdia* and *Cambroraster*. We

have labeled these structures in our figures and added a complementary figure to explain our reasoning.

I. 20 'flattened' here appears to refer to the outline in which case it's not the best descriptor due to the potential for confusion with compression artefacts.

Replaced with linear.

I. 29 But the P elements in *Hurdia* extend well below the margins of the H element as reconstructed by Daley et al. Here (Fig. 5). Here, however, they are enclosed by the H element and it is difficult to interpret their function.

We suspect that the head and trunk of *Hurdia* were quite deep, with the P-elements abutting and enclosing the head ventrolaterally. This is in contrast to the original description in which these elements were reconstructed as suspended anterolateral to the head. These arguments were presented in our paper on *Cambroraster* (cited in this line).

I. 32 'poorly preserved pair of frontal appendages'. Does this affect attempts to interpret the ecology of *Titanokorys*?

It is true that a fine-grained quantitative assessment of prey size is hindered by the appendage preservation quality in *Titanokorys*, and other taxa like *Pahvantia* and *Cordaticaris*. However, the preservation is sufficiently good to recognize ecologically-relevant features, at least at a broad level. For example, the appendages of *Titanokorys* are more similar to *Cambroraster's* than to *Hurdia's*, long endites with a very high number of auxiliary spines, for example. These are also significantly longer compared to *Cambroraster*, a point which we have been able to clarify and better illustrate in our revised figure 3, suggesting a similar microphagous specialization.

p.7, I. 34 'discerned' rather than 'gathered'.

Fixed.

I. 36 They don't look 'needle-like' to me. That would imply a narrower and parallel sided outline.

Replaced with finely tapering.

p. 8, I. 3 The heading and text of Section 3.6 should make it clear that the reinterpretation of *Pahvantia* concerns the appendage morphology (and function), not the identification.

We have renamed the section Morphological Reinterpretation of *Pahvantia hastata*.

p. 12, I. 5-22 This paragraph concludes that carapace morphology does not strongly correlate with interpretations of ecology. This inference is likely correct, but the shortcoming of this approach is that the comparison is with designated ecological categories and not morphological characters of the frontal appendage. The consequence is a number of highly qualified statements such as 'together these observations would seem to speak against', 'it is possible that ... but could also be', 'might have played a part'. While such caution is appropriate, this discussion, and the paper as a whole, would be much stronger if the relationship between carapace characters and appendage characters was explored in a PCA or similar, directly. As it stands the paper does not deliver on its goal of providing a 'broader

quantitative understanding of carapace shape variation and its potential functional and ecological significance’.

We agree that such an approach would permit stronger inferences and we considered various such approaches to get at the relationship between feeding ecology and carapace morphology. Fundamentally, the available data is unfortunately quite limited. Among radiodonts, only *Titanokorys*, *Cambroraster*, *Hurdia*, *Pahvantia*, *Cordaticaris*, *Aegirocassis*, and *Lyrarapax* have both carapaces and appendages complete enough for consideration. Even then, for ca. half of these taxa, precisely quantifying ecologically important variables like the number or spacing of secondary spines is challenging due to overlap of endites, and only single specimens of appendages are known. Even if we were to compare these variables with our carapace shape data, our phylogeny and phylomorphospace results suggests that these ecologically relevant characters would be strongly confounded with phylogeny, which would preclude any confident conclusion. We hope our paper will serve as a starting point for future work, acknowledging the limitations presently imposed by the material available.

Fig. 5 dashed lines are not obvious (or evident?)

Fixed

Reviewer: 2

Comments to the Author(s)

This manuscript by Caron and Moysiuk presents a new radiodont taxon from the Burgess Shale. This is an important fossil locality preserving exceptionally preserved fossils from the Cambrian Explosion, which have proved invaluable for understanding the early evolution of animals. The fossils in this manuscript are beautifully photographed and well described, followed by geometric morphometric analyses and phylogenetic analyses. In general, this is an excellent manuscript that is full of new fossil data and rigorous analyses, and the conclusions of the paper for the most part are well supported by the data. All data and analyses have been provided in the Supplementary Material, and seems to be done to a high standard. I have two major points related to taxonomy and systematics that must be considered before the paper can be published, followed by a series of suggestions for minor to moderate changes. I believe that this paper will be a valuable and interesting contribution to the field once these points have been addressed.

We are thankful for this positive support and for the many helpful comments.

Major comments: Two considerations of Systematics and Taxonomy.

(1) There is no provided justification for *Titanokorys* (replaces *Megaraster* in this MS version) being a new genus. The authors could seriously consider making this taxon a new species of *Cambroraster* rather than a new genus. Although the H-elements are different in size and details of the anatomy, they have an overall broad similarity to the outline. The frontal appendage morphology seems to match that

of *Cambroraster* (except for a possible difference in length of distal endites), and the P-elements are not that different either – in any case both these features are not extremely well known in the new taxon. *Hurdia victoria* and *Hurdia triangulata* have rather different H-elements but were left in the same genus because of the similarity of their other body parts. It is unclear why the authors here consider this taxon to be a new genus, and not just a new species of *Cambroraster*. This must be justified much more clearly and convincingly in the text. Currently the *Titanokorys* diagnosis only includes features of the H-element, and to me this is not enough to justify a new genus. The phylogeny argument (that they are not sister taxa) is weak in this case because the phylogeny is heavily influenced by how the authors code the taxa, and also because there is no resolution between *Titanokorys* and *Cordaticaris*, and *Zhenghecaris* is poorly known. Ultimately the support values for the nodes in this part of the tree are very weak. I think it remains a very real possibility that the new taxon and *Cambroraster* are one genus and two species. This needs to be addressed in the revision.

This was a matter which we gave substantial thought prior to our initial submission. Upon considering all information available to us we decided that assigning a new genus was warranted. We have endeavored to make our justification clearer in the text, as suggested.

Titanokorys and *Cambroraster* differ considerably in terms of the shape of the H-elements (anterior spine, length-width ratio, number of marginal spines, depth of posterior notches) and P-elements (lenticular versus ovoid shape, presence/absence of posterior notches, relative length of the neck region, presence/absence of spines on the necks). They also differ strongly in terms of ornamentation of these elements. It is not possible to include some of these characteristics in the diagnosis because they do not uniquely diagnose *Titanokorys* from other taxa, for example *Cordaticaris* which shares essentially identical carapace ornamentation. In the case of *H. victoria* and *H. triangulata*, the differences are in the H-element outline alone, so we do not consider this a comparable case. In addition, we have added the characters of the secondary spines and terminal endites on podomeres 8 -10 – these spines are much longer compared to any other hurdiid known, including *Cambroraster*.

We note that *Cambroraster* and *Titanokorys* are never found as sister taxa in any of the most parsimonious trees. While the bootstrap support values are relatively weak (as is typical for such a small dataset), and we acknowledge the potential for future discoveries to alter our view, the strict consensus trees provided are the best solution based on the data presently available. The non-sister relationship of *Titanokorys* and *Cambroraster* is robustly recovered under different coding strategies (discrete versus continuous) of H-element shape. Our phylogenetic character coding follows best practices and in our estimation represents the best strategy possible based on the data. We do not see dismissing the phylogenetic results as a justifiable choice in this case.

By the criterion suggested, one could alternatively argue that *Titanokorys*, *Cambroraster*, *Zhenghecaris*, *Cordaticaris*, and perhaps even *Pahvantia* should all be assigned to the same genus. While the delimitation of higher taxa will always be somewhat arbitrary, we think this would be a poor reflection of the morphological diversity of these varied forms.

Finally, the presence of probable new species of *Cambroraster* from North and South China, which plot near the *C. falcatus* cluster in H-element space, further support the delimitation of *Titanokorys* as a separate genus.

(2) The redescription of *Pahvantia* provided by the authors is brilliant, and seems very solid. The information from the counterpart convincingly demonstrates that *Pahvantia* seems to have a very similar frontal appendage to *Hurdia*, as the authors show very clearly in figure 6a-d. I would strongly urge the authors to go even further with this redescription, and consider synonymising *Pahvantia* with *Hurdia*. Given the similarity in setal blades, frontal appendage, and P-element, *Pahvantia hastata* may be best considered as a species of *Hurdia* (ie. *Hurdia hastata*). The difference between *Hurdia victoria* and *Hurdia triangulata* is essentially the different morphology of the H-element, which is also now the case with “*Pahvantia*”. Broadly speaking, the H-elements aren’t that different between “*Pahvantia*” and *Hurdia*. The text needs to include an assessment of what the new interpretation means for the systematics of this taxon (even if you don’t agree with my suggestion to synonymise with *Hurdia*).

We appreciate the supportive comments here. We are reluctant to synonymize *Pahvantia* with *Hurdia* for two main reasons: 1) the goal of this paper was not to redescribe the systematics of *Pahvantia*, and we have not studied all the material available of this taxon (we are aware that new material is being studied by colleagues), although as the reviewer pointed out some similarities exist between the H-element in *Hurdia* and *Pahvantia*; 2) Our phylogenetic result never finds the two as sister taxa. Additionally, although the H-elements are similarly shaped at first impression, our morphometric results show important distinctions, especially in terms of posterior elongation (represented by PC2). The P-elements also differ somewhat broadly in shape (as shown in ESM Figure 6, being more lenticular and having a more robust neck in *Pahvantia* compared to *Hurdia*). Finally, the five elongate endites of *Pahvantia* appear to be roughly equal in length, which differs from the gradual length reduction distally, seen in *Hurdia*. We have added a comment to the manuscript to make our view on this matter clear.

Minor to moderate comments:

Abstract, page 1 line 18: when describing the “variety of ecological niches” only “benthic foragers to agile nektonic apex predators”. Suspension feeding is not listed and could be added.

Suspension feeding was added

Abstract, page 1 lines 26-17: The paper seems to conclude that the variation in shape of the carapace is not really related to feeding ecology, but the abstract here says that the “carapace shape is prone to homoplasy, likely due to a complex and varied relationship with ecology.” This doesn’t seem to reflect correctly the conclusions made in the discussion of the paper.

We rephrased this to clarify our meaning.

Introduction, page 2, line 2: Not all radiodonts have an oral cone strictly speaking, because of taxa such as *Amplectobelua* and *Ramskoeldia* and their GLSs (although these do have toothed plates, just not arranged into an oral cone).

We rephrased this to indicate that radiodonts are typified by the listed characteristics, but they may not be universally present.

Introduction, page 2, line 18: Add reference 1 to the end of the second paragraph, as Daley et al. 2009 were the first to recognise Hurdia as a radiodont and describe it as having a large carapace complex.

Done

Introduction, page 2, lines 19-20: Hurdiids were not known to be radiodonts before 2009. So here when the text says “many described over the past decade” it doesn’t quite correctly describe the history. Nearly all of them were described over the past decade, with the exception of Peytoia, and the concept of hurdiids as a family was only developed within the last decade. Again, reference 1 should be cited here.

We replaced “many” with “the majority”.

Materials and methods, page 3, line 16: correct to “In some case, they appear yellowish...” (“they” was missing)

Fixed.

Materials and methods, page 3 line 24 to page 4 line 2: The description of the geometric morphometrics methods is a bit unclear and too brief here. The landmark analysis methods don’t seem to be complete – there is no statement about what the PCA analysis is based on (presumably the Procrustes coordinates?). On line 30, the Elliptical Fourier Analysis was presumably conducted on outlines, rather than landmarks, but this isn’t actually stated. This text here also gets a bit confusing, because it seems at first reading that the PCA was only conducted on the EFA. It’s only after going through the Supplementary Information (which is also unclear and lacking in places) that I eventually realised that PCA was done for both the landmark and outline datasets. In general, this section of the methods could be expanded in the main text to make a clearer and more detailed description of the morphometrics. The ontogenetic allometry test text doesn’t say which of the datasets (landmark or outline) was used for this part of the methods, and again is very brief, with no real explanation in the supplementary material.

We have expanded the methods section as suggested. All annotated code for the analyses is also available in a supplementary file.

Materials and methods, page 4, lines 3 to 15: The description of the phylogenetic methods are also too brief and quite unclear. The text here introduces two approaches that produce the “D tree” and the “L tree” and the “N tree”, and says that the details are in the supplementary material, however there is no mention of these three trees using the same terminology. This is quite confusing. It is also not adequately explained if the D tree method (five discrete characters) and the L tree method (incorporating species mean shape data) were both done together, or as two separate analyses. If they were done together, it seems like that would be redundant. There are also questions that remain about the details of the methods used, even when reading what was supplied in the supplementary materials. The main text for the D tree method should make it clear that characters from previous analyses were removed, and replaced with new characters in the matrix. The supplementary materials description of the L tree method is unclear. What does it mean to “calculate the mean shape for each species”? What

about the variation in shape seen in each species? Likewise the methods for the phylomorphospace are very brief, and should be expanded for clarity.

We have expanded as recommended and clarified the points made here. Mean shape (i.e. set of mean landmark positions) was calculated with the `mshape` function in `geomorph`, as detailed in the supplementary R code. Early experimentation with several individual specimens per species (to account for polymorphism) produced the same topologies. For ease of presentation, we chose to publish the results based on species mean shapes.

Materials and methods, page 4, lines 16-18: The boxplots were constructed with length measurements only, although elsewhere in the text it mentions that the width of the largest *Cambroraster* specimen exceeds the width of *Titanokorys* (page 10 line 25). The data for width has not been provided. Since this is one of the arguments for these two taxa not being the same species, those width data should be provided and I would suggest include a boxplot for width alongside the boxplot for length in figure 9.

We have included a plot of widths for all taxa, as a compliment to the length data. Any other dimension desired by readers could be easily calculated from the scaled landmark coordinates provided as an ESM file.

Fossil description, page 4, line 19: It would be clearer if this Fossil Description section had two main sections: 3.1 Systematic Paleontology (for *Titanokorys*) and 3.2 Reinterpretation of *Pahvantia hastata*. At the moment it almost reads like *Pahvantia hastata* is included as part of *Megaraster* because the Superphylum, Order, Family headings apply to the entire Fossil Description section including part 3.6.

Agreed, we have made this modification.

Fossil description, page 4, line 25-26: The name *Megaraster* sounds like it is referring to the large size of the rake-like appendage (rather than the large size of the central carapace element). Considering that the appendage is relatively poorly known in *Titanokorys* and it is not included in the diagnosis, maybe a different name could be considered that refers to the large size of the carapace (if the authors decide to keep it as a separate genus, but see my major comment about including it as a new species of *Cambroraster*).

A different name - *Titanokorys* was chosen – see also response to major comments above

Fossil description, pages 5-6: The anatomy of the new taxon morphology is in some places hard to link to the figures provided. When describing anatomical features in the text, refer to the associated acronym labelling the feature in the figure. So for example, when describing ocular notches, refer to “On in figure 1a” instead of just “figure 1a”. Do this throughout the description in order to make it much easier for the reader to link the written description of the specimens.

Yes, we have clarified as suggested.

Fossil description, page 5, lines 22 to 29: This insightful text identifying the anterolateral processes in *Hurdia* seems like it would better fit in the discussion than here.

Yes, we have moved this as suggested.

Fossil description, page 6, lines 32 to 42: Description of the parts of the body other than the carapaces is somewhat lacking in details. The length, shape and paired (or unpaired) nature of the secondary spines of the frontal appendage endites could be described in more detail than just “needle-like”. The reference to “enditic spines” is completely unclear – are these different endites to those already described, are they spines on the endites or different endites themselves... etc. How are they different from the other endites, and are they longer or shorter than the other endites, how many are there, etc. These “enditic spines” looks like they have roughly the same relative length compared to the podomeres as in *Cambroraster*, contrary to what is written in the text.

We have rephrased enditic spines as “distal spiniform endites” and expanded the description. We assume that the secondary spines are unpaired as appears to be typical for hurdiids, but we cannot say with complete confidence based on the material we have. This being said, we have reassessed our material and amended figure 3 to better illustrate the differences between *Cambroraster* and the new taxon. Our new taxon has much longer distal spiniform endites and auxiliary spines (by at least 30%).

Fossil description, page 7, line 39: The authors here interpret the peduncle of *Pahvantia* as just the semi-circular structure, whereas in Lerosey-Aubril & Pates they included the elongated narrow region that extends dorsally upwards from what these authors consider as the base of the peduncle. Some comment as to why the authors do not consider this material to be part of the appendage, contrary to the original description, and what they think this structure is, should be included.

We added a note on this. This material appears more amorphous than the appendage, and was presumably less sclerotized. We suspect it is a part of the larger mass of gill blades and trunk cuticle which overlaps the appendage.

Results Phylogeny, page 9, line 34: The placement of *Schinderhannes* as the most basal hurdiid seems strange given it is so much younger than all the other radiodonts. But I expect that the basal part of the hurdiids clade would be very much changed if the taxon *Ursulinacaris* had been included in the analysis, because of the presence in that hurdiid of paired endites (Pates et al. 2019 *Zoological Letters*).

This phylogenetic position of *Schinderhannes* was recovered previously in the paper describing *Cambroraster*. The characters supporting this grouping can be found in ESM. Of course, late temporal range is not synonymous with derivedness. We see no inherent contradiction in viewing both *Schinderhannes* and *Ursulinacaris* as basal hurdiids. We have discussed these issues further in a different manuscript (in press), which we think is better suited to address them. For now, we have added the reference to *Ursulinacaris*.

Results Phylogeny, page 9, lines 24-27: I think this text here somewhat overstates the helpfulness of carapace characters to the phylogeny. Yes, some of the hurdiids are slightly better resolved, but this comes at the great expense of further destroying resolution further down the tree. Everything other than a hurdiid is just a rake and totally unresolved. This is not ideal. In general, the importance of the phylogenetic results are quite overstated and overanalysed in the discussion (see my further comments below).

We have softened the tone of this section to avoid overstating our point, however -

It is not in fact the addition of carapace characters which result in the noted low resolution, as demonstrated by similarly low resolution in these parts of the tree with no carapace characters included. This appears to be a feature of the dataset owing to conflicting character states among anomalocaridids and ampletobeluids (especially forms like e.g. *Laminacaris*). We note that resolution of the relationships between these forms has been quite low in other recent analyses (e.g. Moysiuk & Caron 2019, Liu et al. 2018, Van Roy et al. 2015). Resolution is somewhat improved using implied weighting – we have included all of these results in the manuscript and supplementary files.

By contrast resolution of hurdiid relationships is finer (and somewhat altered) than had been possible in these previous works (and most relevantly, relative to the previous version of this matrix), thanks to new characters, especially relating to carapace morphology (shape, ornamentation, etc).

Results Phylogeny, page 10, line 6: Refer more clearly to the “phyломorphospace” here rather than the “phylogeny”. There is not overall description of the phyломorphospace results provided here in the results text. Some of the results are presented later in the manuscript, in the discussion (page 11, lines 5-22). A description of the results from both phyломorphospace analyses should be added to the results.

The mistake on line 6 has been corrected. We have included more detail in the result section on the phyломorphospace, at the end of the Phylogeny section, but we feel that the noted lines in the discussion should remain there.

Discussion, page 10, line 13: Here the authors claim that the carapace complex of the new taxon is “significantly” larger than the largest specimens from other Cambrian hurdiids, but no statistical significance text was reported in the results. Also, it was stated elsewhere in the text that the width of some *Cambroraster* specimens is larger than the width of the next taxon.

We have clarified to state that *Titanokorys*’s carapace reaches considerably larger sizes than other taxa. This is best shown in our Figure 9. Because *Cambroraster* has a larger W:L ratio, there is width overlap between the largest specimen and the smallest *Titanokorys*, but *Titanokorys* still achieves larger maximum size (in terms of absolute length or centroid size).

Discussion, page 10, line 20: It is unclear why reference 22 is listed here as giving a size for *Anomalocaris*, when that is not found in the publication. Reference 12 is better for giving the body size of this taxon.

Ref 22 indicates a size of 56cm for *Anomalocaris* (Fig.5). Ref 12 does not provide indication of size.

Discussion, page 10, lines 22-29: I agree with this text that the new taxon and *Cambroraster* are not the same species, however no justification has been provided as to why it needs to be a new genus and not a species of *Cambroraster*. The phylogeny argument (that they are not sister taxa) is weak in this case because the phylogeny is heavily influenced by how the authors code the taxa, and also because there is no resolution between *Titanokorys* and *Cordaticaris*, and *Zhenghecaris* is poorly known. Ultimately the support values for the nodes in this part of the tree are very weak. I think it remains a very real possibility that the new taxon and *Cambroraster* are one genus and two species. This needs to be addressed in the revision.

We address this comment above.

Discussion, page 10 lines 30 to page 11 line 43: The entire discussion about ecology and phylogeny is confusing, circular and difficult to follow. All of this text needs revision. Ultimately, the results show that carapace shape doesn't really have much of an effect on the phylogenetic relationships found in the analysis. This is shown by the phylomorphospace where the things that are closely related on the phylogeny are not plotting near each other in the morphospace. In contrast, feeding ecology (inferred from appendages and nothing to do with carapaces) is quite nicely resolved in the phylogeny. In the discussion text of this paper, the authors seem to confuse and mix up specific feeding ecology derived from interpretation of appendages, with general ecology such as life habit, swimming ability, etc. It goes from saying that there is no "relationship between carapace shape with prey size niche or associated habitat" (page 11 lines 19-20) to saying the opposite that "enlargement of carapaces represent a consequence of adaptation to an increasingly specialized, typically benthic niche linked directly to particular feeding ecology but could also be the side effect of other factors such as the necessity of evolving larger carapaces for defense" (page 11 lines 33-36) and then goes back to saying there is a "lack of evidence for a general correlation between carapace shape and feeding ecology" (page 11 lines 37-38) – this just seems to flip flop back and forth about the link or not between carapace morphology and ecology. This part of the text can probably be made much more concise.

We have revised the text to clarify our positions.

While our results indicate that aspects of carapace shape (e.g. length/width ratio) appear prone to homoplasy, there are localized aspects of signal, for example the shortening of the posterior region of the H-element which unites *Hurdia* spp. relative to other hurdiids.

In this manuscript we mainly discuss feeding ecology, due to the difficulty of testing implications for other aspects of ecology. The referee is correct that lack of a consistent relationship with feeding ecology does not imply lack of ecological significance, which was the point we had intended to get across.

Discussion, page 11, line 23: It is unclear exactly what taxa are meant with "aforementioned hurdiid taxa".

We revised the wording here.

Discussion, page 11, line 26: For the taxa listed as having "smaller carapace complexes" it should be considered that the carapace complex of *Stanleycaris* is not known so we can't say if it is small or large. *Peytoia* has evidence for a carapace that covered all around the front of the head and back as far as the eyes at least. While this doesn't seem to extend forward of the body, it certainly isn't small either. The cephalic carapace situation in *Schinderhannes* is also very poorly known.

We have revised the text here in accordance with these points. While the carapace of *Peytoia* is certainly not as small as, say, *Anomalocaris*, it does not project much beyond the margins of the head as in the taxa which we describe as having truly large carapaces. The carapace elements in *Schinderhannes* may be poorly known, but it seems clear that they are relatively small.

Discussion, page 11, line 27: Is the "more elongate habitus" referring to the appendage or the podomeres or what specifically?

Revised wording.

Discussion, page 11, line 29: The reference here to ref. 2 is not correct, as that paper did not describe anything to do with the three taxa being considered here. Please reference the papers that actually discuss and describe the morphology of these appendages, such as Kuhl et al. 2009; Daley et al. 2009, 2013; Caron et al. 2010, etc.

Corrected.

Discussion, page 11, lines 23 to 32: When discussing the evolutionary trends within hurdiids, the taxon Ursulinacaris is quite relevant and should be included in the discussion (Pates et al. 2019 Zoological Letters).

We added the reference to this paper here. As mentioned above we will be discussing these issues in more detail in a different manuscript, which is currently in press.

Figure 1: The white space between the figure panels could be made a bit thicker to make the distinction between the different photos clearer.

The white space between figure panels has been increased and is now consistent across all figures. We think the increased thickness is adequate and clearly separates each panel figures.

Figure 2: "Fa" is missing from caption.

Corrected.

Figure 3: The label "Sp" is not described in any figure caption. Add a figure panel letter to the drawing and include in the text caption as a separate part of the figure. Move the labels from the photograph of the appendages and oral cone to the drawing of those features. Label the boxes in (a) with the letter label of the closeup in the figure.

This figure has been revised.

Figure 5: It seems unnecessary to have four different angles provided for the reconstruction, when the body is not known and the arrangement of the body parts is derived from other radiodonts. Maybe just a single reconstruction that focuses on the head region would be enough. The rest is just clearly copied from Cambroaster and adds nothing of relevance to the paper.

The different orientations show the other elements of the body like the P-elements and appendages. We feel they are necessary, and not overly space consuming in any case.

Figure 6: There is also a feature labelled as "Sp" in figure 6c. Is this the same feature as the unexplained "Sp" that is found in Figure 3? If not, give them different acronyms.

Corrected. They are not the same so one has been given a new acronym.

Figure 7: Add (a) and (b) for the left and right parts of this figure. In the right figure, label the key taxa Zhenghecaris and Aegirocassis. The purple colour of Aegirocassis is extremely difficult to distinguish from the black dots.

We added rings around Titanokorys.

Figure 8: In (a), remove the “Hurdiinae” line, as this is never discussed or mentioned in the manuscript text.

Corrected.

Appendix B

Associate Editor Comments to Author (Professor Elizabeth Harper):

Thank you for your constructive response to the reviews.

Both reviewers seemingly very happy with the work which has gone into clarifying and improving this manuscript. They both make very minor, but useful comments, and these should be attended to before final acceptance.

Thank you, we appreciate these final positive remarks and have accommodated them, as detailed below.

Reviewer comments to Author:

Reviewer: 1

Comments to the Author(s)

The authors have given the comments of both reviewers careful consideration. Where they do not agree with reviewers' suggested changes or incorporate them fully, the differences are issues of interpretation or emphasis. The authors are entitled to their views, which will be available for testing by future researchers.

The Abstract refers to 'sediment-sifting' and the first paragraph still describes hurdiids as 'specialized for sweep feeding ... in the water column'. In their response the authors explain that the two descriptions are not mutually exclusive but the reference to 'water column' is potentially confusing because it can be construed as eliminating feeding on sediment. This can be easily remedied.

We eliminated the phrase "in the water column" as suggested.

Ms p. 2, l. 9. The suggestion that 'these animals occupied different habitats within the water column' appears to conflict with the possibility of competition.

We are not referring to *Titanokorys* here. Whether it may occupy a similar niche to *Cambroraster* is discussed later. Prior work convincingly demonstrates ecological differences between other previously described species which would limit competition.

Ms p. 2, l. 27. 'together' is redundant.

Corrected.

Ms p. 3, l. 11. 'preserved as aluminosilicate and carbon films' requires a citation.

Citation added.

Ms p. 5, l. 30. Explain what 'buried flat' means?

We rephrased this sentence to clarify.

Ms p. 6, l. 22. 'specimen' is redundant here and elsewhere. In this case 'holotype' is adequate.

We eliminated “specimen” in all redundant cases.

Ms p. 8, l. 17. I suggest ‘compatible’ rather than ‘in tune’.

Replaced with “consistent”.

Ms p. 11, l. 18. reminiscent ‘of’ not ‘to’

Corrected.

Ms p. 12, l. 29. elevates ‘to’ not ‘as’

We rephrased this sentence.

Reviewer: 2

Comments to the Author(s)

In the revised version of this manuscript, the authors have adequately addressed all the reviewers comments. The authors have added a lot of new images and drawings of the material, which better illustrate their descriptive work and create a clearer and more coherent foundation for their taxonomic decisions. They also clarified areas of the text to do with the methods and results, making the paper much easier to understand. In general, this revised version is very much improved over the previous version, which was already excellent. I have just a few very minor suggestions below, and I recommend accepting this manuscript once these have been addressed.

Minor suggestions:

With all the changes to the figures, it seems that some call-outs to figures in the text don’t accurately correspond anymore with the appropriate image. For example, on page 9 lines 39-40, the text about the holotype appendages and oral cones refers to figure 3b, but this is actually a photo of just the carapace – the authors probably mean to call-out to figure 3a or maybe 3g? All call-outs should be double-checked that they refer to the correct figure.

Thank you for catching this, we have checked and corrected all issues with callouts.

In figure 3, I suggest that the authors place all labels of anatomy in the drawing in figure 3k, and remove the labels from the associated photograph in figure g. This allows the reader to see more clearly the fossil, without labels getting in the way.

We have moved most labels to the diagram as suggested.

The authors have expanded a bit the statement about a deformed carapace, and figured the original and retrodeformed specimen in figures 3b and 3c. The arrows are not mentioned in the caption, but they seem to point in the opposite direction that the shear force and deformation. I guess maybe the arrows are showing the direction of retrodeformation, rather than the deformation direction itself? It would

make more sense for the arrows to be reversed and show the direction of warping, and also then mention in the caption clearly what the arrows represent.

We reversed the arrows and added a statement in the caption to clarify.

Also on this subject, in the text on page 7 line 27, and page 3 line 32, the authors attribute this deformation to oblique burial of the specimen relative to bedding, however it looks much more like a metamorphic deformation. This is because the agnostids found in the same image are also deformed, with those aligned roughly horizontally looking too flat/thin and those aligned roughly vertical looking too wide/fat. This suggests the whole surface underwent some kind of slight metamorphic warping/deformation, which makes sense given this is in a mountainous region.

After detailed consideration we do not think that metamorphic shearing is a major factor here. Not all of the agnostids are deformed consistently with a single plane of shearing. This would suggest that burial orientation has differentially affected them too. Those agnostids that share a similar deformation pattern to the H-element are buried below the H-element and may have been oriented parallel to it. The other *Titanokorys* body elements on the slab also lack signs of a shared pattern of shearing with the H-element.

On page 7 line 45, the use of the words “finely tapering” to replace some previous text is a bit vague. Does “finely” mean that the taper proceeds gradually along the entire length of the spine, or does it mean that it only tapers right at the end? Replace with a clearer terminology.

Replaced with “gradually tapering”.